# Operator-based quantum thermodynamic uncertainty relations

Pratik Sathe,[*] Luis Pedro García-Pintos,[†] and Francesco Caravelli[‡]

*Theoretical Division (T-4), Los Alamos National Laboratory, New Mexico, 87545, USA*

(Dated: July 29, 2024)

The Heisenberg uncertainty relation, which links the uncertainties of the position and momentum of a particle, has an important footprint on the quantum behavior of a physical system. Motivated by this principle, we propose that thermodynamic currents associated with work, heat, and internal energy are described by well-defined Hermitian operators; i.e., we associate physical observables to quantum thermodynamic flows. The observables are defined such that their expectation values match the average values of the associated currents. These rates, or currents, differ from their classical counterparts due to the non-commutativity of the corresponding operators. Using the Robertson-Schrödinger uncertainty relation, we then obtain various thermodynamic uncertainty relationships between them. In particular, we connect the fluctuations in heat rate and thermodynamic power with those in internal energy. We further illustrate this approach by applying it to quantum batteries, where we derive an energy-power uncertainty relationship and show how measurements affect the fluctuations.

## I. INTRODUCTION

Quantum thermodynamics is an emerging field of physics that aims to understand the non-equilibrium behavior of small quantum systems [1, 2]. One of the successes of this program is the derivation and refinement of the laws of thermodynamics by formalizing notions of heat, work and entropy for out-of-equilibrium open quantum systems [3, 4]. For example, refinements of the second law of thermodynamics, in the form of work fluctuation equalities were first proven in the classical setting [5–8], and were soon followed by the derivations of their quantum mechanical counterparts [9–11] (see Refs. [12, 13] for reviews).

In recent years, another related set of results, referred to as 'thermodynamic uncertainty relations' (TURs), were proven to constrain processes in classical stochastic thermodynamics [14]. These relations are a direct consequence of irreversibility and connect fluctuations with dissipation. They assert that the precision of thermodynamic currents in non-equilibrium steady states (NESSs) is bounded from below by the inverse of the entropy production [15–22]. If $J$ denotes a thermodynamic current and $\Sigma$ is the average entropy production, the TURs typically assume the form

$$\frac{\sigma_J^2}{\langle J \rangle^2} \geq \frac{2}{\Sigma}, \tag{1}$$

where $\langle \circ \rangle$ and $\sigma_\circ^2 = \langle \circ^2 \rangle - \langle \circ \rangle^2$ denote the mean and variance of a quantity $\circ$ respectively. Variants of Eq. (1) where its right-hand side is replaced by a function of $\Sigma$ have also been derived [18, 21].

Understanding the quantum mechanical analog of TURs is a topic of active research that has attracted

significant attention. Quantum TURs and associated relations have been derived for quantum clocks [23], for NESSs in continuous Markovian quantum dynamics [24–27], for discrete quantum Markovian dynamics using optimal transport theory [28], for periodic slow driving [29, 30] and for general open quantum systems [31].

In this paper, we present a new type of TUR, which we call operator-based TURs, that is based on the non-commutativity of thermodynamic rate operators. Similar to the Heisenberg uncertainty principle that relates the uncertainties in position and momentum, we obtain uncertainty relations between pairs of thermodynamic quantities using the Robertson-Schrödinger uncertainty relation [32]. The obtained relations state that the product of the variance of two thermodynamic quantities must be lower bounded by their commutator, thus capturing a uniquely quantum property. These bounds assume a form different from conventional TURs [Eq. (1)], stating instead that

$$\sigma_{\boldsymbol{O}_1}^2 \sigma_{\boldsymbol{O}_2}^2 \geq f(\boldsymbol{O}_1, \boldsymbol{O}_2, \boldsymbol{\rho}), \tag{2}$$

where $\boldsymbol{O}_{1,2}$ denote two operators and $\rho$ denotes the density matrix of system and the environment. Here, $f$ is a function that involves the commutator and the covariance of the two operators. (A visual representation of the type of relationships we derive is shown in Fig. 1.) This approach thus provides a recipe to relate the uncertainties of various thermodynamic currents as well as of thermodynamic quantities (such as internal energy) that can be represented as quantum mechanical observables.

Since work and heat are process-dependent quantities, as opposed to being state functions [33], they cannot be represented as quantum mechanical observables [34] (see also, Refs. [35–37]). Instead, work is usually obtained using a two-point measurement procedure which is accompanied by a loss of coherence [10, 38–40]. However, their associated currents, such as the work rate (i.e., power), the heat flow, and the internal energy rate can be represented as observables with associated Hermitian operators. While it is known that the average values of

---

[*] psathe@lanl.gov

[†] lpgp@lanl.gov

[‡] caravelli@lanl.gov

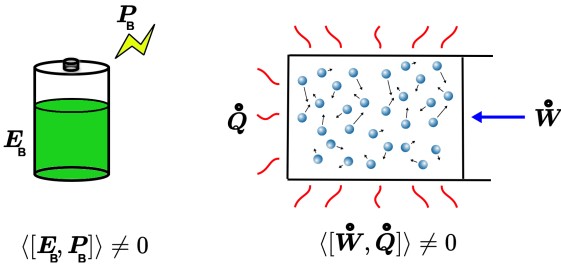

FIG. 1. A pictorial representation of the non-commutativity of the energy ($\boldsymbol{E}_B$) and power ($\boldsymbol{P}_B$) operators for quantum batteries, and the power ($\overset{\circ}{\boldsymbol{\mathcal{W}}}$) and heat flow ($\overset{\circ}{\boldsymbol{\mathcal{Q}}}$) operators in an open system. The precision of measurements of $\boldsymbol{E}_B$ and $\boldsymbol{P}_B$ for the battery (or $\overset{\circ}{\boldsymbol{\mathcal{W}}}$ or $\overset{\circ}{\boldsymbol{\mathcal{Q}}}$ for more general open quantum dynamics) are lower bounded by the degree of non-commutativity of these operators.

these currents equal the expectation values of the corresponding operators [41–46], particularly in the case of weak system-bath coupling, the operators themselves and their higher moments or fluctuations have not been studied to our knowledge. (See, however, Refs. [45, 46], which study the fluctuations of *work* due to continuous power measurements, and Ref. [47], which examines work fluctuations using integrated power.) We derive expressions for these operators for the cases of general open quantum systems as well as those whose dynamics are governed by a Markovian master equation. Due to the generality of the Robertson-Schrödinger uncertainty relations, unlike conventional TURs which are usually applicable only to NESSs, our expressions are applicable at any stage in the evolution of a quantum system, and not just for steady states.

Using various examples of open and closed quantum systems, we derive expressions for power, heat flow and energy rate operators, and numerically and analytically compare their fluctuations with the constraints implied by the operator-based uncertainty bounds. We further highlight the utility of this approach by applying it to quantum batteries, which exploit quantum states and phenomena such as superposition and entanglement as opposed to classical batteries, which store and release energy through electro-chemical reactions. Quantum batteries leverage the principles of quantum mechanics to create highly efficient energy storage systems (see Ref. [48] for a recent review), and hold the promise of a quantum advantage in terms of faster charging and work-extraction [49–55] compared to their classical analogues, and have also been realized experimentally [56, 57].

Quantum batteries thus serve as an interesting and novel platform for exploring quantum thermodynamic phenomena. We establish energy-power uncertainty relationships for both open and closed quantum batteries by properly defining battery energy and battery power operators. These relations can be interpreted to mean that the uncertainty in battery energy is inversely re-

lated to the uncertainty in battery charging power. We also derive the average or typical uncertainty expected in various scenarios by Haar-averaging across the state of the system.

The rest of the paper is organized as follows: In Sec. II, we discuss the notation, introduce average heat and work rates in quantum thermodynamics, and review the Robertson-Schrödinger uncertainty relation. In Sec. III, we present operator representations of heat and work rates, followed by derivations of various operator-based TURs in Sec. IV. In Sec. V, we turn to quantum batteries and derive energy-power uncertainty relations. In addition, we use the Weingarten calculus to derive typical uncertainty values in Sec. VI, for the uncertainty relations derived in the rest of the paper. We conclude with a summary of our results and an outlook in Sec. VII.

## II. PRELIMINARY DISCUSSION

### A. Notation and Setup

We will denote quantum operators by bold capital letters, while classical variables will be denoted by capital letters in standard font. The expectation value of an observable $\boldsymbol{A}$ will be denoted by $\langle \boldsymbol{A} \rangle = \mathrm{Tr}(\boldsymbol{\rho}\boldsymbol{A})$ with $\boldsymbol{\rho}$ denoting the system-environment density matrix. We will consider the cases of general open quantum dynamics described by a system-environment Hamiltonian, open Markovian quantum systems evolving according to a Lindblad equation as well as closed but not isolated quantum systems.

Let us first discuss the most general case of an open quantum system, i.e., one described by a Hamiltonian that determines the evolution of the system-environment composite system:

$$\boldsymbol{H}_{\mathrm{tot}}(t) = \boldsymbol{H}_S(t) \otimes \mathbb{I}_E + \mathbb{I}_S \otimes \boldsymbol{H}_E + \boldsymbol{V}_{SE}. \quad (3)$$

Here, the system Hamiltonian $\boldsymbol{H}_S$ operates only on the system Hilbert space $\mathcal{H}_S$, and the environment Hamiltonian $\boldsymbol{H}_B$ operates only on the environment Hilbert space $\mathcal{H}_E$. The system-environment interaction $\boldsymbol{V}_{SE}$ acts on the full Hilbert space $\mathcal{H}_S \otimes \mathcal{H}_E$. The density matrix of the composite system will be denoted by $\boldsymbol{\rho}_{SE}$, while the reduced system density matrix will be denoted by $\boldsymbol{\rho}_S = \mathrm{Tr}_E(\boldsymbol{\rho})$, i.e., the partial trace of the full density matrix with respect to the environment.

The average internal energy of the system is

$$E(t) = \mathrm{Tr}\{\boldsymbol{H}_S \boldsymbol{\rho}_{SE}\} = \mathrm{Tr}_S\{\boldsymbol{H}_S \boldsymbol{\rho}_S\}. \quad (4)$$

Taking the first derivative with respect to time, we obtain

$$\dot{E}(t) = \mathrm{Tr}_S\{\dot{\boldsymbol{H}}_S \boldsymbol{\rho}_S\} + \mathrm{Tr}_S\{\boldsymbol{H}_S \dot{\boldsymbol{\rho}}_S\} \quad (5)$$

The two terms on the right hand side of Eq. (5) are usually identified as the rate of change of work (i.e., power)

and of heat flow respectively [4]:

$$\dot{W}(t) = \text{Tr}_S\{\dot{\boldsymbol{H}}_S\boldsymbol{\rho}_S\}, \qquad (6a)$$

$$\dot{Q}(t) = \text{Tr}_S\{\boldsymbol{H}_S\dot{\boldsymbol{\rho}}_S\}. \qquad (6b)$$

Thus, the non-equilibrium quantum first law of thermodynamics assumes the familiar form:

$$\dot{E}(t) = \dot{W}(t) + \dot{Q}(t). \qquad (7)$$

In the case of a quantum battery, the battery energy is defined with respect to a time-independent reference Hamiltonian, called the battery Hamiltonian, which we will denote in this paper by $\boldsymbol{H}_0$. In the case of an open quantum battery, we regard the battery as being the system, with the rest being the environment. In that case, $\boldsymbol{H}_0$ operates only on $\mathcal{H}_S$. The battery energy $E_B(t)$ is then given by

$$E_B(t) := \langle \boldsymbol{H}_0 \rangle = \text{Tr}_S(\boldsymbol{H}_0\boldsymbol{\rho}_S(t)). \qquad (8)$$

The reduced density matrix $\rho_S(t)$ evolves according to the Liouville-von Neumann equation with the full system-environment Hamiltonian, or according to a Lindblad master equation in the case of open quantum Markovian dynamics.

The instantaneous charging power of a battery is also defined with respect to $\boldsymbol{H}_0$, so that

$$P_B(t) := \frac{\text{d}E_B(t)}{\text{d}t} \qquad (9a)$$

$$= \text{Tr}_S(\boldsymbol{H}_0\dot{\boldsymbol{\rho}}_S) \qquad (9b)$$

We note that the battery energy $E_B(t)$ and battery charging power $P_B(t)$ are distinct from thermodynamic internal energy $E(t)$ from Eq. (4) and thermodynamic power $\dot{W}(t)$, since the latter are defined with respect to the full system Hamiltonian $\boldsymbol{H}_S$. We will discuss these subtleties in more detail in Sec. V.

**B. The Robertson-Schrödinger uncertainty relation**

To derive the operator-based TURs, we will rely on the Robertson-Schrödinger uncertainty relationship [32, 58, 59]. It states that, for any Hermitian operators $\boldsymbol{A}$ and $\boldsymbol{B}$,

$$\sigma_{\boldsymbol{A}}^2\sigma_{\boldsymbol{B}}^2 \geq \frac{1}{4}|\langle[\boldsymbol{A},\boldsymbol{B}]\rangle|^2 + |\text{cov}(\boldsymbol{A},\boldsymbol{B})|^2, \qquad (10)$$

where

$$\text{cov}(\boldsymbol{A},\boldsymbol{B}) := \frac{1}{2}\langle\{\boldsymbol{A},\boldsymbol{B}\}\rangle - \langle\boldsymbol{A}\rangle\langle\boldsymbol{B}\rangle \qquad (11)$$

is the covariance of the two operators, and $\sigma_{\boldsymbol{A}}^2 = \text{cov}(\boldsymbol{A},\boldsymbol{A})$ is an operator's variance. Here, $[\boldsymbol{A},\boldsymbol{B}] = \boldsymbol{A}\boldsymbol{B} - \boldsymbol{B}\boldsymbol{A}$ and $\{\boldsymbol{A},\boldsymbol{B}\} = \boldsymbol{A}\boldsymbol{B} + \boldsymbol{B}\boldsymbol{A}$ are the commutator and anti-commutator of the two operators, respectively.

We note that the Robertson uncertainty relation is given by Eq. (10), but with only the commutator term on the right hand side.

The most well-known application of this uncertainty relationship (obtained from retaining only the commutator term) is Heisenberg's uncertainty principle, which relates the uncertainties between the momentum and position of a particle: $\sigma_{\boldsymbol{x}}\sigma_{\boldsymbol{p}} \geq \hbar/2$.

**III. OPERATOR REPRESENTATION OF THERMODYNAMIC FLOWS**

In this section, we discuss the operator representations of thermodynamic power, heat flow, and the rate of change of internal energy of a quantum system.

**A. Thermodynamic Flow Operators From System-Environment Hamiltonians**

Consider an open quantum system described by a time-dependent Hamiltonian with a system Hamiltonian $\boldsymbol{H}_S(t)$ and an environment Hamiltonian $\boldsymbol{H}_E$. If their interaction is mediated by an interaction term $\boldsymbol{V}_{SE}$, the dynamics of the joint state $\boldsymbol{\rho}_{SE}(t)$ of $S$ and $E$ is determined by the total Hamiltonian

$$\boldsymbol{H}_{\text{tot}}(t) = \boldsymbol{H}_S(t) \otimes \mathbb{I}_E + \mathbb{I}_S \otimes \boldsymbol{H}_E + \boldsymbol{V}_{SE}. \qquad (12)$$

(Often, only the system Hamiltonian $\boldsymbol{H}_S$ is time-dependent, an assumption we make here. However, all our definitions and operator-based TUR derivations are valid even with time-dependent $\boldsymbol{V}_{SE}$ and $\boldsymbol{H}_E$.)

The system's average energy is then given by

$$E(t) = \langle \boldsymbol{H}_S(t) \rangle = \text{Tr}_S\left(\boldsymbol{H}_S(t)\boldsymbol{\rho}_S(t)\right), \qquad (13)$$

The rate at which the system's energy changes can be written as

$$\dot{E}(t) = \dot{W}(t) + \dot{Q}(t), \qquad (14)$$

where [recalling Eq. (6)],

$$\dot{W}(t) = \text{Tr}_S\left(\dot{\boldsymbol{H}}_S(t)\boldsymbol{\rho}_S(t)\right), \qquad (15a)$$

$$\dot{Q}(t) = \text{Tr}_S\left(\boldsymbol{H}_S(t)\dot{\boldsymbol{\rho}}_S(t)\right). \qquad (15b)$$

are usually interpreted as the power and heat flow [4].

This leads to natural definitions of the *heat flow operator* $\dot{\boldsymbol{\mathcal{Q}}}$ and the *power operator* $\dot{\boldsymbol{\mathcal{W}}}$, defined by the property that the average heat rate and average power are equal to the expectation values of the corresponding operators. That is, we impose that $\dot{\boldsymbol{\mathcal{Q}}}$ and $\dot{\boldsymbol{\mathcal{W}}}$ satisfy

$$\dot{W}(t) = \langle \dot{\boldsymbol{\mathcal{W}}}(t) \rangle = \text{Tr}_S(\dot{\boldsymbol{\mathcal{W}}}(t)\boldsymbol{\rho}_S(t)), \qquad (16a)$$

$$\text{and } \dot{Q}(t) = \langle \dot{\boldsymbol{\mathcal{Q}}}(t) \rangle = \text{Tr}_{SE}(\dot{\boldsymbol{\mathcal{Q}}}(t)\boldsymbol{\rho}_S(t)). \qquad (16b)$$

For a system described by a Hamiltonian of the form (3), the operators

$$\mathring{\mathcal{W}}(t) = \dot{\boldsymbol{H}}_S(t), \tag{17a}$$

$$\text{and } \mathring{\mathcal{Q}}(t) = \frac{-i}{\hbar}[\boldsymbol{H}_S(t), \boldsymbol{V}_{SE}], \tag{17b}$$

satisfy Eqs. (16a) and (16b). For completeness, we also define an internal energy rate operator $\mathring{\mathcal{U}}$ as

$$\mathring{\mathcal{U}} = \mathring{\mathcal{W}} + \mathring{\mathcal{Q}}. \tag{18}$$

We emphasize that $\mathring{\mathcal{U}}$ is in general not the time derivative of the internal energy operator $\boldsymbol{U}$.

Equation (17a) for $\mathring{\mathcal{W}}(t)$ follows from comparing Eq. (15a) and Eq. (16a). To derive (17b) for $\mathring{\mathcal{Q}}(t)$, we use the Liouville–von Neumann equation and Eq. (15b), which leads to

$$\dot{Q}(t) = \text{Tr}_S\left(\boldsymbol{H}_S \text{Tr}_E(\dot{\boldsymbol{\rho}}_{SE})\right) \tag{19a}$$

$$= -\frac{i}{\hbar}\text{Tr}_{SE}\left(\boldsymbol{H}_S\left[\boldsymbol{H}_{\text{tot}}, \boldsymbol{\rho}_{SE}\right]\right) \tag{19b}$$

$$= \text{Tr}_{SE}\left\{\frac{-i}{\hbar}[\boldsymbol{H}_S, \boldsymbol{H}_{\text{tot}}]\boldsymbol{\rho}_{SE}\right\} \tag{19c}$$

$$= \text{Tr}_{SE}\{\mathring{\mathcal{Q}}(t)\boldsymbol{\rho}_{SE}\} = \langle\mathring{\mathcal{Q}}(t)\rangle, \tag{19d}$$

which leads us to the definition Eq. (17b) of $\mathring{\mathcal{Q}}(t)$.

We remind the reader that power and heat flows have different units compared to the heat and work, as they represent work and heat transfer per unit of time. We also note that, while the $\mathring{\mathcal{W}}$ and $\mathring{\mathcal{Q}}$ correspond to work rate and heat rate respectively, they are not, in general, time derivatives of some unspecified work and heat operators.

## B. Thermodynamic Flow Operators From Lindblad Master Equations

These ideas can also be applied to the case where an open quantum system's dynamics are described by a Lindblad master equation. Consider a Lindblad master equation

$$\frac{\text{d}\boldsymbol{\rho}}{\text{d}t} = -\frac{i}{\hbar}[\boldsymbol{H}_S(t), \boldsymbol{\rho}] + \mathcal{D}_t[\boldsymbol{\rho}], \tag{20a}$$

$$\text{with } \mathcal{D}_t[\circ] = \sum_k \gamma_k(t)\left(\boldsymbol{L}_k \circ \boldsymbol{L}_k^\dagger - \frac{1}{2}\{\boldsymbol{L}_k^\dagger\boldsymbol{L}_k, \circ\}\right), \tag{20b}$$

where $\gamma_k(t) > 0$. Here, the $\boldsymbol{L}_k$s denote the Lindblad operators and $\mathcal{D}_t$ denotes the possibly-time dependent dissipator, which includes all contributions to the non-unitary evolution of $\boldsymbol{\rho}$ [60, 61].

Using Eq. (6) and the cyclicity of trace, we observe that

$$\mathring{\mathcal{W}} \equiv \dot{\boldsymbol{H}}_S, \tag{21a}$$

$$\text{while } \mathring{\mathcal{Q}} = \mathcal{D}_t^*[\boldsymbol{H}_S] \tag{21b}$$

$$\text{where } \mathcal{D}_t^*[\circ] := \sum_k \gamma_k(t)\left(\boldsymbol{L}_k^\dagger \circ \boldsymbol{L}_k - \frac{1}{2}\{\boldsymbol{L}_k^\dagger\boldsymbol{L}_k, \circ\}\right) \tag{21c}$$

Thus, the heat flow operator is equal to a slight modification $\mathcal{D}^*$ of the dissipator $\mathcal{D}$, acting upon the system Hamiltonian $\boldsymbol{H}_S$.

We can use the same formalism to study other thermodynamic quantities. For instance, the rate of change of the von Neumann entropy of the system is:

$$\dot{S} = -k_B\text{Tr}(\dot{\boldsymbol{\rho}}\log\boldsymbol{\rho}), \tag{22a}$$

$$= -k_B\text{Tr}(\boldsymbol{\rho}\mathcal{D}_t^*[\log\boldsymbol{\rho}]) \tag{22b}$$

$$= \langle\mathring{\mathcal{S}}_t[\boldsymbol{\rho}]\rangle \tag{22c}$$

$$\text{where } \mathring{\mathcal{S}}_t[\circ] = -k_B\mathcal{D}_t^*[\log\circ]. \tag{22d}$$

Thus, we see that while heat and work rates can be described by operators, the entropy rate can instead be described by an entropy rate *superoperator* $\mathring{\mathcal{S}}_t$.

## C. Examples

Next, we use the expressions for $\mathring{\mathcal{Q}}$ and $\mathring{\mathcal{W}}$ to study flows in a few illustrative examples. For all of the numerical simulations in this paper, we use the Python software package QuTip [62, 63]. For the two examples presented below, we discuss numerical simulations in Appendix A and show numerically that, as expected, $\frac{\text{d}\langle\boldsymbol{U}\rangle}{\text{d}t} = \langle\mathring{\mathcal{U}}\rangle$.

### 1. Two interacting spins

Let us consider two interacting spin-half particles. In this simple model, we consider one of the particles to be the system with the other one serving as its environment. Specifically, we consider a composite system with a Hamiltonian

$$\begin{aligned} \boldsymbol{H}_{\text{tot}} &= \boldsymbol{H}_S \otimes \mathbb{I} + \boldsymbol{V}_{SE} + \mathbb{I} \otimes \boldsymbol{H}_E \\ \text{with } \boldsymbol{H}_S &= f(t)\,\boldsymbol{\sigma}^x \\ \boldsymbol{V}_{SE} &= g\,\boldsymbol{\sigma}^z \otimes \boldsymbol{\sigma}^z \\ \boldsymbol{H}_E &= \boldsymbol{\sigma}^x, \end{aligned} \tag{23}$$

where $f(t)$ is a time-dependent function. Plugging into (17b) and (17a), we obtain:

$$\mathring{\mathcal{Q}} = -\frac{2}{\hbar}f(t)g\boldsymbol{\sigma}^y \otimes \boldsymbol{\sigma}^z \tag{24a}$$

$$\mathring{\mathcal{W}} = \dot{f}(t)\boldsymbol{\sigma}^x \otimes \mathbb{I}, \tag{24b}$$

$$\text{and } \mathring{\mathcal{U}} = \mathring{\mathcal{W}} + \mathring{\mathcal{Q}}. \tag{24c}$$

These operators thus assume experimentally accessible forms, and could potentially be accessed on a gate-based quantum computer.

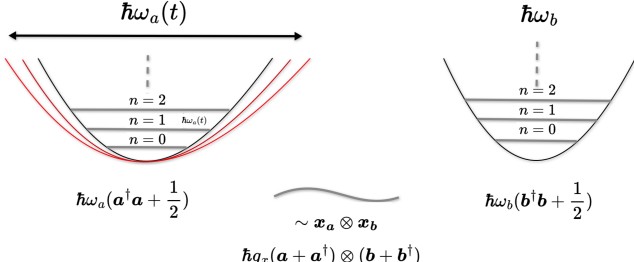

FIG. 2. Pictorial representation of a system of two interacting one-dimensional harmonic oscillators [see Eq. (25)] . The first oscillator (left) is regarded as the system, with the second one as the environment. Work is done to the system by changing the frequency $\omega_a$ as a function of time and heat is exchanged by the system with the second oscillator due to the position-position coupling.

### 2. Two interacting oscillators

We now consider another simple example of two one-dimensional harmonic oscillators coupled to each other (represented pictorially in Fig. 2), with one serving as the system and the other as the environment. Specifically, we consider [64–66]:

$$\boldsymbol{H}_{\text{tot}} = \boldsymbol{H}_S \otimes \mathbb{I} + \boldsymbol{V}_{SE} + \mathbb{I} \otimes \boldsymbol{H}_E$$
$$\text{with } \boldsymbol{H}_S = \frac{\boldsymbol{p}_a^2}{2m} + \frac{1}{2}m\omega_a(t)^2\boldsymbol{x}_a^2,$$
$$\boldsymbol{H}_E = \frac{\boldsymbol{p}_b^2}{2m} + \frac{1}{2}m\omega_b^2\boldsymbol{x}_b^2,$$
$$\text{and } \boldsymbol{V}_{SE} = 2g\boldsymbol{x}_a \otimes \boldsymbol{x}_b. \qquad (25)$$

The position and momentum operators for the first oscillator (i.e., the system) are denoted by $\boldsymbol{x}_a$ and $\boldsymbol{p}_a$ respectively, while those corresponding to the second oscillator (which serves as the environment) are denoted by $\boldsymbol{x}_b$ and $\boldsymbol{p}_b$ respectively.

The power and heat flow operators are then easily obtained using Eq. (17):

$$\mathring{\boldsymbol{\mathcal{W}}} = m\omega_a\dot{\omega}_a\boldsymbol{x}_a^2 \otimes \mathbb{I}_b; \qquad (26a)$$

$$\mathring{\boldsymbol{\mathcal{Q}}} = \frac{-2g}{m}\boldsymbol{p}_a \otimes \boldsymbol{x}_b. \qquad (26b)$$

Note that $\mathring{\boldsymbol{\mathcal{W}}}$ is explicitly time dependent given that $\omega_a$ is time dependent. Furthermore, $\mathring{\boldsymbol{\mathcal{W}}}$ is proportional to the harmonic oscillator potential energy, $\mathring{\boldsymbol{\mathcal{W}}} \propto \boldsymbol{x}_a^2$.

## IV. THERMODYNAMIC UNCERTAINTY RELATIONS

In the previous section, we proposed definitions for heat flow and power operators for any open quantum system. For some simple examples, we found that these operators assume simple, experimentally accessible forms [see Eqs. (24) and (26)].

An immediate consequence of the definitions of flow operators is that the fluctuations in power and fluctuations in the heat rate are related to each other by

$$\sigma_{\mathring{\boldsymbol{\mathcal{W}}}}^2\sigma_{\mathring{\boldsymbol{\mathcal{Q}}}}^2 \geq \frac{1}{4}\left|\langle[\mathring{\boldsymbol{\mathcal{Q}}},\mathring{\boldsymbol{\mathcal{W}}}]\rangle\right|^2 + \left|\text{cov}(\mathring{\boldsymbol{\mathcal{Q}}},\mathring{\boldsymbol{\mathcal{W}}})\right|^2. \qquad (27)$$

That is, an uncertainty relation exists between the fluctuations in work and heat flows. The proof of Eq. (27) follows from the Robertson-Schrödinger uncertainty relation applied to $\mathring{\boldsymbol{\mathcal{Q}}}$ and $\mathring{\boldsymbol{\mathcal{W}}}$.

We observe from Eq. (4) that the internal energy operator $\boldsymbol{U} = \boldsymbol{H}_S$. Thus, $\mathring{\boldsymbol{\mathcal{W}}}$ and $\mathring{\boldsymbol{\mathcal{Q}}}$ satisfy uncertainty relations with the internal energy and its change rate, too, which we expand upon next.

### A. Internal energy uncertainty relations

We now derive a range of bounds on the internal energy fluctuations of open quantum systems. First, we recall that $\mathring{\boldsymbol{\mathcal{U}}} = \mathring{\boldsymbol{\mathcal{W}}} + \mathring{\boldsymbol{\mathcal{Q}}}$, defined in Eq. (18) is the operator corresponding to the rate of change of internal energy. Generally, $[\boldsymbol{U},\mathring{\boldsymbol{\mathcal{U}}}] \neq 0$, and consequently, using the Robertson-Schrödinger uncertainty relation Eq. (10), we first obtain our first bound, which is

$$\sigma_{\boldsymbol{U}}^2 \geq \frac{\frac{1}{4}\left|\langle[\boldsymbol{U},\mathring{\boldsymbol{\mathcal{U}}}]\rangle\right|^2 + \left|\text{cov}(\boldsymbol{U},\mathring{\boldsymbol{\mathcal{U}}})\right|^2}{\sigma_{\mathring{\boldsymbol{\mathcal{U}}}}^2} \qquad (28a)$$

$$\geq \frac{\frac{1}{4}\left|\langle[\boldsymbol{U},\mathring{\boldsymbol{\mathcal{U}}}]\rangle\right|^2 + \left|\text{cov}(\boldsymbol{U},\mathring{\boldsymbol{\mathcal{U}}})\right|^2}{(\sigma_{\mathring{\boldsymbol{\mathcal{Q}}}} + \sigma_{\mathring{\boldsymbol{\mathcal{W}}}})^2 - 2t_-}, \qquad (28b)$$

Here, $t_\pm$ is defined as

$$t_\pm = \sqrt{\frac{1}{4}\left|\langle[\mathring{\boldsymbol{\mathcal{Q}}},\mathring{\boldsymbol{\mathcal{W}}}]\rangle\right|^2 + \left|\text{cov}(\mathring{\boldsymbol{\mathcal{Q}}},\mathring{\boldsymbol{\mathcal{W}}})\right|^2}$$
$$\pm \text{cov}(\mathring{\boldsymbol{\mathcal{Q}}},\mathring{\boldsymbol{\mathcal{W}}}), \qquad (29)$$

In order to derive Eq. (28b), we used the fact that the uncertainties of $\mathring{\boldsymbol{\mathcal{U}}}$ are related to those of $\mathring{\boldsymbol{\mathcal{W}}}$ and $\mathring{\boldsymbol{\mathcal{Q}}}$ as $\sigma_{\mathring{\boldsymbol{\mathcal{U}}}}^2 \leq (\sigma_{\mathring{\boldsymbol{\mathcal{Q}}}} + \sigma_{\mathring{\boldsymbol{\mathcal{W}}}})^2 - 2t_-$. (For a proof, see Appendix. B.)

Similarly, an application of the Robertson-Schrödinger uncertainty relation to the pairs $(\boldsymbol{U},\mathring{\boldsymbol{\mathcal{Q}}})$ and $(\boldsymbol{U},\mathring{\boldsymbol{\mathcal{W}}})$ gives us

$$\sigma_{\boldsymbol{U}}^2 \geq \frac{\frac{1}{4}\left|\langle[\boldsymbol{U},\mathring{\boldsymbol{\mathcal{Q}}}]\rangle\right|^2 + \left|\text{cov}(\boldsymbol{U},\mathring{\boldsymbol{\mathcal{Q}}})\right|^2}{\sigma_{\mathring{\boldsymbol{\mathcal{Q}}}}^2}, \qquad (30a)$$

$$\text{and } \sigma_{\boldsymbol{U}}^2 \geq \frac{\frac{1}{4}\left|\langle[\boldsymbol{U},\mathring{\boldsymbol{\mathcal{W}}}]\rangle\right|^2 + \left|\text{cov}(\boldsymbol{U},\mathring{\boldsymbol{\mathcal{W}}})\right|^2}{\sigma_{\mathring{\boldsymbol{\mathcal{W}}}}^2}. \qquad (30b)$$

To our knowledge, our approach as well as the specific uncertainty relations we derived have not been appreciated in the literature. It is worth noting that while Ref. [67] employs Robertson-Schrödinger uncertainty relations, the conclusions therein are substantially different and are applicable primarily to open Gaussian systems.

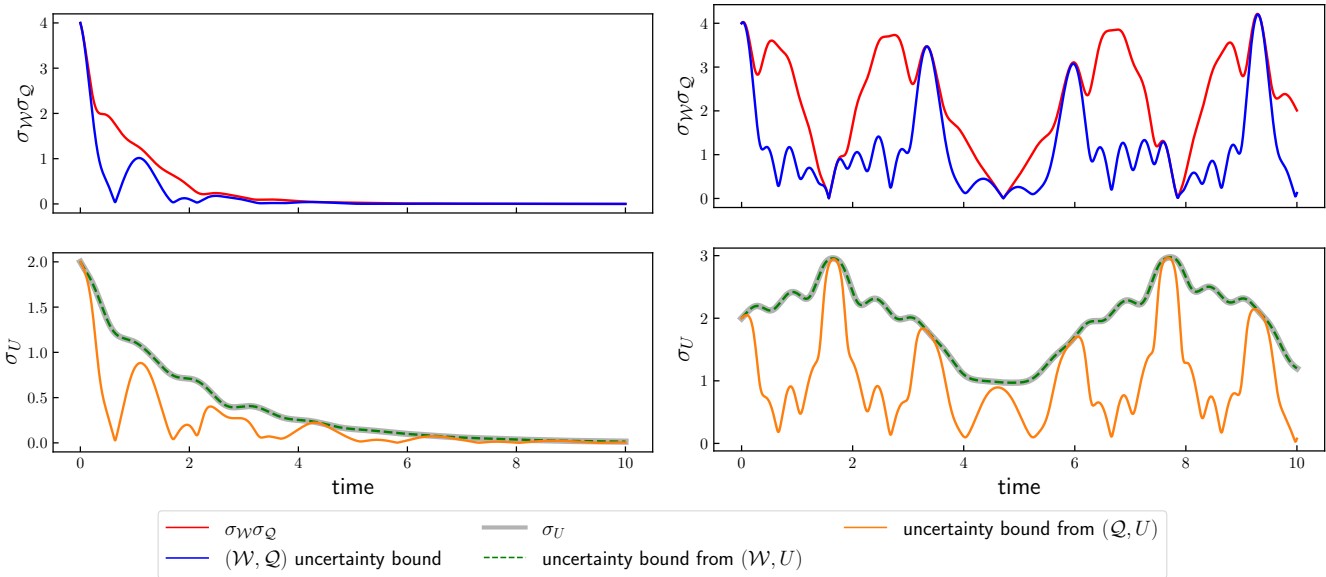

FIG. 3. Numerically computed operator variances ($\sigma_{\mathring{\mathcal{W}}}, \sigma_{\mathring{\mathcal{Q}}}$ and $\sigma_U$) and the corresponding lower bounds obtained using the Robertson-Schrödinger uncertainty relation for a system of two interacting spins with a Hamiltonian (23). The left panel corresponds to $f(t) = 2\exp(-t/2)$, while the right panel corresponds to $f(t) = \sin(t) + 2$, with $\hbar = g = 1$ in both cases. The upper panel shows numerically computed $\sigma_{\mathring{\mathcal{W}}}\sigma_{\mathring{\mathcal{Q}}}$ as well as the lower bound from Eq. (33a). The lower panels show $\sigma_U$ as well as lower bounds on it obtained using Eq. (30a) and Eq. (30b). In all cases, the initial state was chosen to be $|\psi(0)\rangle = |\uparrow\rangle \otimes |\uparrow\rangle$, with the up ket representing the $\boldsymbol{\sigma}^z$ eigenvectors corresponding to the eigenvalue $+1$. We note that the curve for $\sigma_U$ coincides with the bound obtained using Eq. (30b).

## B. Examples

We now numerically verify the various bounds on $\sigma_U$ as well as on $\sigma_{\mathring{\mathcal{Q}}}\sigma_{\mathring{\mathcal{W}}}$ for two examples below.

### 1. Two interacting qubits

Returning to the example of two interacting driven spins described by the Hamiltonian (23), plugging in the expressions for the power and heat rate operators (24) into (27), we see that the commutator and covariance terms simplify to

$$\left| \frac{1}{2} \langle \{\mathring{\mathcal{Q}}, \mathring{\mathcal{W}}\} \rangle - \langle \mathring{\mathcal{Q}} \rangle \langle \mathring{\mathcal{W}} \rangle \right|^2 = h(t)^2 \tag{31}$$
$$\times |\langle \boldsymbol{\sigma}^y \otimes \boldsymbol{\sigma}^z \rangle|^2 |\langle \boldsymbol{\sigma}^x \otimes \mathbb{I} \rangle|^2,$$
$$\text{and } \left| \frac{1}{2i} \langle [\mathring{\mathcal{Q}}, \mathring{\mathcal{W}}] \rangle \right|^2 = h(t)^2 |\langle \boldsymbol{V}_{SE} \rangle|^2, \tag{32}$$

where $h(t) = \frac{2gf(t)\dot{f}(t)}{\hbar}$. From Eq. (27), we thus have

$$\sigma_{\mathring{\mathcal{Q}}}\sigma_{\mathring{\mathcal{W}}} \geq \frac{g}{\hbar} \frac{\mathrm{d}f^2(t)}{\mathrm{d}t} \sqrt{\langle \boldsymbol{V}_{SE} \rangle^2 + \langle \boldsymbol{\sigma}^y \otimes \boldsymbol{\sigma}^z \rangle^2 \langle \boldsymbol{\sigma}^x \otimes \mathbb{I} \rangle^2} \tag{33a}$$

$$\geq \frac{g}{\hbar} \frac{\mathrm{d}f^2(t)}{\mathrm{d}t} \langle \boldsymbol{V}_{SE} \rangle, \tag{33b}$$

where, the bound in the second line is simply the application of the Robertson uncertainty relation in which only the commutator term is retained. Thus, the power-heat rate uncertainty is lower bounded by the average system-environment interaction.

Numerical simulations of this system for two choices of the driving function $f(t)$ are shown in Fig. 3. We compare the values of $\sigma_{\mathring{\mathcal{Q}}}\sigma_{\mathring{\mathcal{W}}}$ as well as those of $\sigma_U$ with the corresponding lower bounds obtained from Eqs. (33a), (30a) and (30b).

Note that for both choices of $f(t)$, $\sigma_U$ is tightly bound by (30b). In this example, the bound is saturated simply because $\boldsymbol{U} \propto \mathring{\mathcal{W}}$ with a time-dependent proportionality factor, as seen by comparing Eq. (23) and Eq. (24b). Consequently, in the derivation of the Robertson-Schrödinger uncertainty relation, where the Cauchy-Schwarz inequality is used, we get an equality instead of an inequality. Furthermore, since the two operators commute, the uncertainty is fully captured by the covariance term.

### 2. Two interacting oscillators

For two interacting oscillators with a Hamiltonian given by Eq. (25), the power-heat rate uncertainty re-

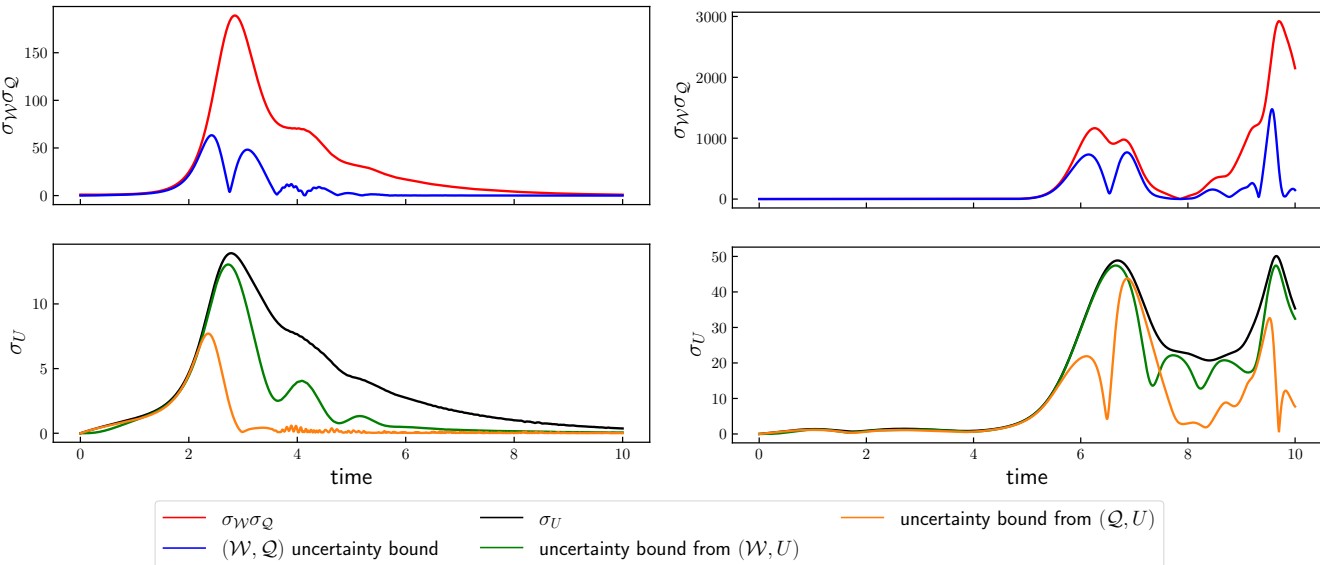

FIG. 4. Numerically computed operator variances ($\sigma_{\hat{\mathcal{W}}}, \sigma_{\hat{\mathcal{Q}}}$ and $\sigma_U$) and the corresponding lower bounds obtained using the Robertson-Schrödinger uncertainty relation for a system of two driven, interacting oscillators with a Hamiltonian (25). The left panel corresponds to $\omega_a(t) = 2 \exp(-t/2)$, while the right panel corresponds to $\omega_a(t) = \sin(t) + 2$, with $\hbar = g = m = \omega_b = 1$ in both cases. The upper panel shows numerically computed $\sigma_{\hat{\mathcal{W}}} \sigma_{\hat{\mathcal{Q}}}$ as well as the lower bound from Eq. (34a). The lower panels show $\sigma_U$ as well as lower bounds on it obtained using Eq. (30a) and Eq. (30b). In all cases, the initial state was chosen to be $|\psi(0)\rangle = |0\rangle \otimes |0\rangle$, i.e. the tensor product of the ground states of the two oscillators.

lation Eq. (27) becomes

$$\sigma_{\hat{\mathcal{Q}}}^2 \sigma_{\hat{\mathcal{W}}}^2 \geq \hbar^2 (\omega_a \dot{\omega}_a)^2 |\langle \boldsymbol{V}_{SE} \rangle|^2 + (\omega_a \dot{\omega}_a)^2$$
$$\times \left| 2g\langle \boldsymbol{x}_a^2 \rangle \langle \boldsymbol{p}_a \boldsymbol{x}_b \rangle + i\hbar \langle \boldsymbol{V}_{SE} \rangle - \langle \boldsymbol{p}_a \boldsymbol{x}_a \boldsymbol{V}_{SE} \rangle \right|^2 \tag{34a}$$

$$\geq \hbar^2 (\omega_a \dot{\omega}_a)^2 |\langle \boldsymbol{V}_{SE} \rangle|^2. \tag{34b}$$

Once again, the lower bound in the Robertson uncertainty relation is proportional to the rate of change of the system frequency and the average system-environment interaction.

In Fig. 4, we numerically compute the values of $\sigma_{\hat{\mathcal{W}}} \sigma_{\hat{\mathcal{Q}}}$ and $\sigma_{\boldsymbol{U}}$ and compare them against the uncertainty bounds derived above, for two different choices of $\omega_a(t)$. Unlike the case of two interacting spins, none of the bounds is saturated identically (except at specific times), since none of the operators under consideration commute with each other.

## V. QUANTUM BATTERIES: ENERGY-POWER UNCERTAINTY RELATIONS

In this section, we discuss operatorial representations for battery energy and battery power, and derive uncertainty relations between them.

The energy stored in a quantum battery is quantified with resepct to an operator $\boldsymbol{H}_0$, also sometimes known as the battery Hamiltonian, which describes its energy levels. The battery energy and the instantaneous battery

power are measured against $\boldsymbol{H}_0$, as already discussed in Sec. II A.

### A. Closed quantum batteries

Let us consider the case of a closed quantum battery evolving unitarily. We will define an energy operator $\boldsymbol{E}_B$ and a power operator $\boldsymbol{P}_B$ whose expectation values characterize the average energy and power of the battery, respectively. Consider a battery self-Hamiltonian $\boldsymbol{H}_0$ and an additional, potentially time-dependent charging potential $\boldsymbol{V}_S(t)$, so that the full Hamiltonian is

$$\boldsymbol{H}_{\text{tot}}(t) = \boldsymbol{H}_S(t) = \boldsymbol{H}_0 + \boldsymbol{V}_S(t). \tag{35}$$

The charging potential's role is to drive the battery to a state with a higher energy. It is assumed that $\boldsymbol{V}_S(t)$ is zero before and after the charging protocol.

Since $E_B(t) = \text{Tr}(\boldsymbol{H}_0 \boldsymbol{\rho}(t))$ [recall Eq. (8)], we identify $\boldsymbol{H}_0$ as being the battery energy operator $\boldsymbol{E}_B$:

$$\boldsymbol{E}_B = \boldsymbol{H}_0. \tag{36}$$

We define the battery's power operator as

$$\boldsymbol{P}_B^c := -\frac{i}{\hbar}[\boldsymbol{H}_0, \boldsymbol{V}_S]. \tag{37}$$

(The superscript $c$ denotes that the system is closed.) $\boldsymbol{P}_B^c$ is a Hermitian operator, since both $\boldsymbol{H}_0$ and $\boldsymbol{V}_S$ are Hermitian operators. Clearly, if $[\boldsymbol{H}_0, \boldsymbol{V}_S] = 0$, $E(t) =$

$E(0)$ for all $t$, so we $[\boldsymbol{H}_0, \boldsymbol{V}_S] \neq 0$ during the charging process of the battery. It is straightforward to show that $P_B(t) = \langle \boldsymbol{P}_B^c \rangle$, with $P_B(t)$ defined in Eq. (9b). The steps involved are similar to those in Eq. (19).

The following uncertainty relationship follows from Eq. (10):

$$\sigma_{\boldsymbol{E}_B}^2 \sigma_{\boldsymbol{P}_B^c}^2 \geq \frac{1}{4\hbar^2} |\langle [\boldsymbol{H}_0, [\boldsymbol{H}_0, \boldsymbol{V}_S]] \rangle|^2 + \left| \mathrm{cov}(\boldsymbol{H}_0, \frac{-i}{\hbar}[\boldsymbol{H}_0, \boldsymbol{V}_S]) \right|^2. \tag{38}$$

This uncertainty relationship characterizes a deviation between classical and quantum batteries.

One may expect that the uncertainty of the energy should reduce with a reduction in the uncertainty of power, since the average value of the latter equals the derivative of the average value of the former. However, from Eq. (38), we conclude that this intuition is wrong, for the same reasons that it is wrong for the position and momentum of a quantum particle. Equation (38) complements other uncertainty relations that constrain quantum batteries. References [54, 68] prove trade-off relations between energy and extractable work fluctuations and a battery's average charging power.

Let us note that despite superficial similarities, thermodynamic power from previous sections is different from battery power considered here. First, we note that the battery energy operator $\boldsymbol{E}_B$ is different from the thermodynamic internal energy operator $\boldsymbol{U}$. Unless $\boldsymbol{V}_S = 0$, we have $\boldsymbol{E}_B = \boldsymbol{H}_0 \neq \boldsymbol{H}_S = \boldsymbol{U}$. Furthermore, we note that the average instantaneous battery power $P_B$ from (9b) is different from the average instantaneous power $\dot{W}(t)$ from Eq. (6a). Since we are considering a closed quantum system, there is no heat exchange so that $\dot{W}(t)$ also equals the rate of change of the thermodynamic energy $P(t) := \dot{E}(t)$.

While the instantaneous quantities differ, the change in the average internal energy is equal to the change in the battery energy over a charging protocol, as long as $\boldsymbol{V}_S(t_i) = \boldsymbol{V}_S(t_f) = 0$ where $t_i$ and $t_f$ denote the starting and ending times of the charging protocol. Denoting the change in internal energy by $\Delta E \equiv E(t_f) - E(t_i)$ and the change in the battery energy by $\Delta E_B \equiv E_B(t_f) - E_B(t_i)$, we have

$$\Delta E = E(t_f) - E(t_i) \tag{39a}$$
$$= \mathrm{Tr}[\boldsymbol{H}_S(t_f)\boldsymbol{\rho}_S(t_f)] - \mathrm{Tr}[\boldsymbol{H}_S(t_i)\boldsymbol{\rho}_S(t_i)] \tag{39b}$$
$$= \mathrm{Tr}[\boldsymbol{H}_0\boldsymbol{\rho}(t_f)] - \mathrm{Tr}[\boldsymbol{H}_0\boldsymbol{\rho}(t_i)] \tag{39c}$$
$$= E_B(t_f) - E_B(t_i) = \Delta E_B. \tag{39d}$$

Alternatively, we may write

$$(\Delta E =) \int_{t_i}^{t_f} \mathrm{d}t P(t) = \int_{t_i}^{t_f} \mathrm{d}t P_B(t) \ (= \Delta E_B) \tag{40a}$$
$$\text{or} \int_{t_i}^{t_f} \mathrm{d}t \langle \mathring{\boldsymbol{\mathcal{U}}}(t) \rangle = \int_{t_i}^{t_f} \mathrm{d}t \langle \boldsymbol{P}_B(t) \rangle \tag{40b}$$

Thus, while the instantaneous rate of change of internal energy is different from the instantaneous rate of change of the battery energy, their integrals over a charging protocol are equal.

We note that the change in the battery energy is equivalent to the notion of 'exclusive work' [5] done on a system. In contrast, we used the notion of 'inclusive work' in our definitions of thermodynamic work. In short, exclusive work refers to the change in energy with respect to a fixed term in the system Hamiltonian, while inclusive work is computed with respect to the full system Hamiltonian. We refer the reader to Refs. [13, 69] for more detailed discussions of inclusive vs exclusive work.

For completeness, we note that the thermodynamic power and heat flow operators for Eq. (35) are

$$\mathring{\boldsymbol{\mathcal{W}}} = \frac{\mathrm{d}\boldsymbol{H}(t)}{\mathrm{d}t} = \dot{\boldsymbol{V}}_S(t) \tag{41a}$$
$$\mathring{\boldsymbol{\mathcal{Q}}} = -\frac{i}{\hbar}[\boldsymbol{H}(t), \boldsymbol{H}(t)] = 0, \tag{41b}$$

where $\mathring{\boldsymbol{\mathcal{Q}}} = 0$ since the system is closed.

## B. Open quantum batteries

Next, we consider two descriptions of open quantum batteries. In one, a battery couples to an environment (which can include reservoirs and ancillae) with a total Hamiltonian:

$$\boldsymbol{H}_{\mathrm{tot}}(t) = \boldsymbol{H}_S \otimes \mathbb{I} + \boldsymbol{V}_{SE} + \mathbb{I} \otimes \boldsymbol{H}_E. \tag{42}$$

In the second case, the battery dynamics is described by a Lindblad master equation. In both cases, we find that the battery power operator can be split into a closed battery power operator, and one due to the interaction with the environment.

Consider the system-environment Hamiltonian in Eq. (42). The system Hamiltonian is again split into a battery Hamiltonian and a charging potential as

$$\boldsymbol{H}_S(t) = \boldsymbol{H}_0 + \boldsymbol{V}_S(t). \tag{43}$$

While the battery energy operator is $\boldsymbol{E}_B \equiv \boldsymbol{H}_0$, the battery power operator $\boldsymbol{P}_B$ is

$$\boldsymbol{P}_B = \boldsymbol{P}_B^c \otimes \mathbb{I} + \boldsymbol{P}_B^o \tag{44a}$$
$$\text{with} \ \boldsymbol{P}_B^c = -\frac{i}{\hbar}[\boldsymbol{H}_0, \boldsymbol{V}_S(t)], \tag{44b}$$
$$\text{and} \ \boldsymbol{P}_B^o = -\frac{i}{\hbar}[\boldsymbol{H}_0 \otimes \mathbb{I}, \boldsymbol{V}_{SE}]. \tag{44c}$$

Here, the closed quantum battery operator $\boldsymbol{P}_B^c$ is identical to the one obtained previously [Eq. (37)]. The interaction with the environment contributes an additional term $\boldsymbol{P}_B^o$, due to the non-commutativity of the battery Hamiltonian with the system-environment interaction.

It holds that

$$\frac{\mathrm{d}P_B(t)}{\mathrm{d}t} \equiv \mathrm{Tr}([\boldsymbol{H}_0 \otimes \mathbb{I}]\dot{\boldsymbol{\rho}}(t)) = \langle \boldsymbol{P}_B \rangle. \qquad (45)$$

As in the case of a closed quantum battery, we expect that $[\boldsymbol{E}_B, \boldsymbol{P}_B] \neq 0$, so that the energy-power uncertainty relation, in general, should include a non-zero contribution due to the commutator of the energy and power operators.

Next, consider the case of an open quantum battery described by a Lindblad master equation (20a). The system Hamiltonian $\boldsymbol{H}_S$ is split into a sum of a battery Hamiltonian and a charging potential, same as in Eq. (43).

The average battery power simplifies as follows:

$$P_B(t) = \frac{\mathrm{d}E_B(t)}{\mathrm{d}t} = \mathrm{Tr}[\boldsymbol{H}_0 \dot{\boldsymbol{\rho}}(t)] \qquad (46a)$$

$$= \mathrm{Tr}[\boldsymbol{P}_B \rho] \qquad (46b)$$

$$\text{where } \boldsymbol{P}_B = \boldsymbol{P}_B^c + \boldsymbol{P}_B^o, \qquad (46c)$$

$$\text{with } \boldsymbol{P}_B^o = \mathcal{D}_t^*[\boldsymbol{H}_0], \qquad (46d)$$

with $\mathcal{D}_t^*$ given in Eq. (21c). $\boldsymbol{P}_B^c$ is the closed quantum battery power operator that was derived in the context of closed quantum batteries, while $\boldsymbol{P}_B^o$ is the contribution due to dissipation.

We note that the dissipated battery power $\boldsymbol{P}_B^o$ is related, but in general different from the heat flow operator from Eq. (21b), since

$$\mathring{\mathcal{Q}} = \mathcal{D}_t^*[\boldsymbol{H}_S] = \boldsymbol{P}_B^o + \mathcal{D}_t^*[\boldsymbol{V}_S(t)]. \qquad (47)$$

Applying the Robertson-Schrödinger uncertainty relations to the $\boldsymbol{E}_B$ and $\boldsymbol{P}_B$ operators, we can thus obtain a lower bound on $\sigma_{\boldsymbol{E}_B}\sigma_{\boldsymbol{P}_B}$:

$$\sigma_{\boldsymbol{P}_B}^2 \sigma_{\boldsymbol{E}_B}^2 \geq \frac{1}{4}|\langle[\boldsymbol{P}_B, \boldsymbol{E}_B]\rangle|^2 + |\mathrm{cov}(\boldsymbol{P}_B, \boldsymbol{E}_B)|^2 \qquad (48)$$

## C. Effects of measurements and decoherence, and maximum uncertainty bounds

For all the uncertainty relations derived above, the predominantly quantum contribution is due to the non-commutativity of the operators being considered. While the covariance term also is quantum mechanical, it ceases to become so when the operators commute. Thus, when the two operators commute, the only non-zero contribution comes from the covariance term, which simply reduces to the covariance of two random variables.

Hence, it is natural to consider the effects of decoherence on the bound, and especially on the commutator term in the uncertainty relations. First, we note that in the Robertson-Schrödinger uncertainty relations for two operators $\boldsymbol{A}$ and $\boldsymbol{B}$, the commutator term involves the expectation value with respect to the state of the system. For convenience, we define

$$\sqrt{\mathcal{B}} = \frac{1}{2}|\mathrm{Tr}([\boldsymbol{A}, \boldsymbol{B}]\boldsymbol{\rho})|, \qquad (49)$$

so that the Robertson uncertainty relation states that $\sigma_{\boldsymbol{A}}^2 \sigma_{\boldsymbol{B}}^2 \geq \mathcal{B}$.

Let us denote the spectral projectors of $\boldsymbol{A}$ and $\boldsymbol{B}$ by sets of orthogonal projectors $\{\boldsymbol{\Pi}_A^i\}$ and $\{\boldsymbol{\Pi}_B^i\}$ respectively. Thus, we have $[\boldsymbol{A}, \boldsymbol{\Pi}_A^i] = 0$, $[\boldsymbol{B}, \boldsymbol{\Pi}_B^i] = 0$, $\sum_i \boldsymbol{\Pi}_*^i = \mathbb{I}$ and $(\boldsymbol{\Pi}_*^i)^2 = \boldsymbol{\Pi}_*^i$. We now define two operations,

$$\mathcal{D}_*(\cdot) = \sum_i \boldsymbol{\Pi}_*^i \cdot \boldsymbol{\Pi}_*^i, \qquad (50a)$$

$$\text{and } \mathcal{C}_*(\cdot) = (\cdot) - \sum_i \boldsymbol{\Pi}_*^i \cdot \boldsymbol{\Pi}_*^i, \qquad (50b)$$

which select the diagonal and off-diagonal elements of any operator respectively in the eigenbasis of $*$. Then, we can write

$$\boldsymbol{\rho} = \mathcal{C}_A(\boldsymbol{\rho}) + \mathcal{D}_A(\boldsymbol{\rho}) = \mathcal{C}_B(\boldsymbol{\rho}) + \mathcal{D}_B(\boldsymbol{\rho}). \qquad (51)$$

Using the definitions above, it follows that

$$2\sqrt{\mathcal{B}} = |\mathrm{Tr}([\boldsymbol{A}, \mathcal{C}_A(\boldsymbol{\rho})]\boldsymbol{B})| = |\mathrm{Tr}([\boldsymbol{B}, \mathcal{C}_B(\boldsymbol{\rho})]\boldsymbol{A})|, \qquad (52)$$

i.e., the non-commutativity of the operators $\boldsymbol{A}, \boldsymbol{B}$ depends only on the off-diagonal elements of the density matrix in the basis of $\boldsymbol{A}$ and $\boldsymbol{B}$ respectively. For the case of battery energy-power uncertainty relations, the operator pairs of interest are the energy and power operators. Measurements in the basis of the energy and power will immediately imply that post-measurement, the quantity $\mathcal{B}$ is zero. More generally, greater decoherence result in a smaller value of $\mathcal{B}$. (The same holds true for other uncertainty relations, such as the power-heat flow operator TUR.)

Let us now study the maximum value attainable by $\sqrt{\mathcal{B}}$. First, we note that

$$2\sqrt{\mathcal{B}} = |\mathrm{Tr}([\boldsymbol{A}, \boldsymbol{\rho}]\boldsymbol{B})| = |\mathrm{Tr}([\boldsymbol{B}, \boldsymbol{\rho}]\boldsymbol{A})|. \qquad (53)$$

Using the Cauchy-Schwarz inequality, we can obtain two upper bounds on the commutator term, which can be combined in to the expression

$$\mathcal{B} \leq \min\left(\frac{1}{4}\|[\boldsymbol{A}, \mathcal{C}_A(\boldsymbol{\rho})\|_F^2\|\boldsymbol{B}\|_F^2, \frac{1}{4}\|[\boldsymbol{B}, \mathcal{C}_B(\boldsymbol{\rho})\|_F^2\|\boldsymbol{A}\|_F^2\right), \qquad (54)$$

where $\|\boldsymbol{C}\|_F^2 = \mathrm{Tr}(\boldsymbol{C}^\dagger \boldsymbol{C})$ is the Frobenius matrix norm of a matrix $\boldsymbol{C}$, which also equals the sum of squares of its eigenvalues if $\boldsymbol{C}$ is Hermitian. Since $\boldsymbol{\rho}^\dagger = \boldsymbol{\rho}$, we have $\mathrm{Tr}(\boldsymbol{\rho}^2) = \|\boldsymbol{\rho}\|_F^2 = \mathcal{P}$, where $\mathcal{P}$ denotes the purity of $\boldsymbol{\rho}$. Thus, if $\boldsymbol{\rho}$ is diagonal either in the basis of $\boldsymbol{A}$ or $\boldsymbol{B}$, the quantum uncertainty can go to zero.

We also note that the commutator term can be written in terms of the coherence in terms of the $l_2$-induced norm [70] in the basis of $\boldsymbol{B}$:

$$\mathbb{C}_B(\boldsymbol{A}) = \|\mathcal{C}_B(\boldsymbol{A}))\|_F^2 = \frac{1}{2}\sum_i \left\|[\boldsymbol{A}, \boldsymbol{\Pi}_j^B]\right\|_F^2. \qquad (55)$$

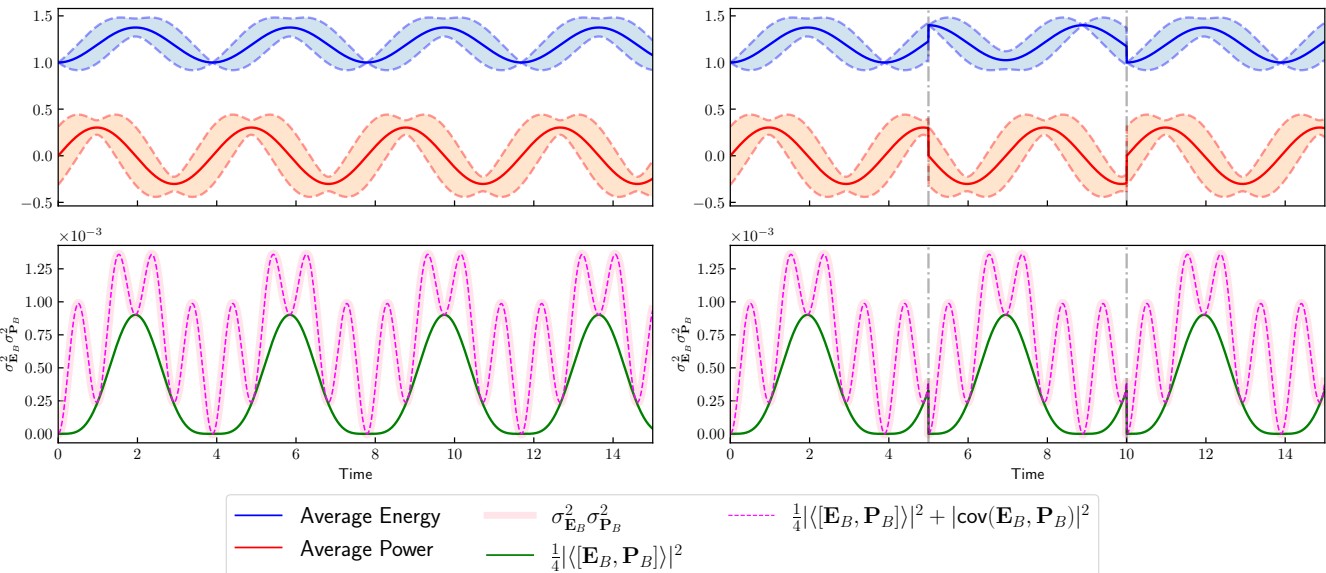

FIG. 5. Energy-power uncertainty for a two-level system described by (59) without (left figure) and with two equally-spaced measurements in the $\sigma^3$ basis (right figure). Upper panel: Evolution of average energy and average power are shown in solid lines (blue and red respectively), and the width of the shaded region represents the uncertainty $2\sigma_{E_B}$ and $2\sigma_{P_B}$. Specifically, we plot $\langle E_B \rangle$ sandwiched between $\langle E_B \rangle \pm \sigma_{E_B}$ and similarly for $P_B$. Lower panel: The product $\sigma_{P_B}^2 \sigma_{E_B}^2$ is shown along with the lower bound (38). We show the commutator term separately, along with the bound that includes the covariance term. The simulations correspond to the values $h_0 = 1.2$, $h_3 = 0.2$, $v_0 = 0$ and $\vec{v} = (0.5, 0.6, 0)$. The initial condition $\rho(t = 0)$ was chosen by setting $\vec{\beta} = (0, 0, 0.5)$ in (60), i.e. the initial condition corresponds to the spin-down state in the $\boldsymbol{\sigma}^z$ basis. We set $\hbar = 1$ for convenience.

Using Lemma 1 from Ref. [71], we have the following bound:

$$\|[\boldsymbol{A}, \boldsymbol{B}]\|_F^2 \leq 4\|\boldsymbol{A}\|_F^2 \mathbb{C}_A(\boldsymbol{B}) \tag{56}$$

We will now use this bound in order to obtain an upper bound on the commutator term for various examples.

#### 1. Closed quantum battery

In the case of the energy-power uncertainty relation for a closed quantum battery, $\boldsymbol{A} = \boldsymbol{H_0}$ and $\boldsymbol{B} = \boldsymbol{P}_B^c = -\frac{i}{\hbar}[\boldsymbol{H}_0, \boldsymbol{V}_S]$. We then obtain,

$$\mathcal{B} \leq \frac{1}{4\hbar^2}\|[\boldsymbol{H}_0, \mathcal{C}_{H_0}(\boldsymbol{\rho}_S)]\|_F^2 \|[\boldsymbol{H}_0, \boldsymbol{V}_S]\|_F^2 \tag{57a}$$

$$\leq \frac{4}{\hbar^2}\|\boldsymbol{H}_0\|_F^4 \mathbb{C}_{H_0}(\boldsymbol{\rho}_S) \mathbb{C}_{H_0}(\boldsymbol{V}_S) \tag{57b}$$

which shows that the uncertainty is upper bounded by both the $l_2$ induced-norm coherence of the interaction potential and the density matrix.

#### 2. Heat and work rates

Similar bounds can be obtained for the uncertainty for an open system's heat flux and work flow defined in (17b)

and (17a) respectively. We obtain

$$\mathcal{B} \leq \frac{4}{\hbar^2}\|\boldsymbol{H}_S\|_F^4 \mathbb{C}_{H_S}(\boldsymbol{\rho}_{SE}) \mathbb{C}_{H_S}(\boldsymbol{V}_{SE}) \tag{58}$$

which also shows that the uncertainty is zero if either $\boldsymbol{\rho}$ or $\boldsymbol{V}_{SE}$ are diagonal in the basis of the subsystem.

### D. Examples

#### 1. Closed quantum system: A single qubit

Consider a qubit with a total Hamiltonian

$$\begin{aligned} \boldsymbol{H}_{\text{tot}} &= \boldsymbol{H}_0 + \boldsymbol{V}_S(t) \\ \text{with } \boldsymbol{H}_0 &= h_0 \boldsymbol{\sigma}^0 + h_3 \boldsymbol{\sigma}^z \\ \text{and } \boldsymbol{V}_S(t) &= (v_0 \boldsymbol{\sigma}^0 + \vec{v}.\vec{\boldsymbol{\sigma}})\theta(t), \end{aligned} \tag{59}$$

where $h_0, h_3, v_1, v_2, v_3 \in \mathbb{R}$, and $\theta(t)$ denotes the Heaviside step function. For $t \geq 0$, we have

$$\boldsymbol{H}_{\text{tot}} = \boldsymbol{H}_S = \alpha_0 \boldsymbol{\sigma}^0 + \vec{\alpha} \cdot \vec{\boldsymbol{\sigma}},$$

where $\alpha_0 = h_0 + v_0, \alpha_1 = v_1, \alpha_2 = v_2, \alpha_3 = h_3 + v_3$, and $\vec{\sigma} \equiv \begin{pmatrix} \boldsymbol{\sigma}^x & \boldsymbol{\sigma}^z & \boldsymbol{\sigma}^z \end{pmatrix}$.

The time evolution operator at time $t > 0$ evaluates to

$$
\begin{aligned}
\boldsymbol{U}(0 \to t) &= \mathcal{T} \exp\left(-i \int_0^t \boldsymbol{H}(t') \mathrm{d}t'\right) \\
&= \exp\left(-i(\alpha_0 \boldsymbol{\sigma}^0 + \vec{\alpha} \cdot \vec{\boldsymbol{\sigma}})t\right) \\
&= e^{-i\alpha_0 t} e^{-it(\vec{\boldsymbol{\sigma}} \cdot \vec{\alpha})} \\
&= e^{-i\alpha_0 t} \left( \boldsymbol{\sigma}^0 \cos(t\alpha) - i\frac{\sin(t\alpha)}{\alpha}(\vec{\boldsymbol{\sigma}} . \vec{\alpha}) \right),
\end{aligned}
$$

The density matrix $\boldsymbol{\rho}_0$ at $t = 0$ can be written as

$$
\boldsymbol{\rho}_0 = \frac{1}{2}\left(\boldsymbol{\sigma}^0 + \vec{\beta} \cdot \vec{\boldsymbol{\sigma}}\right), \tag{60}
$$

where $\beta \equiv \left|\vec{\beta}\right| \leq 1$ is required for $\boldsymbol{\rho}_0$ to be a valid density matrix [72]. The purity is $\mathcal{P} = \frac{1+|\vec{\beta}|^2}{2}$.

The density matrix at time $t > 0$ is

$$
\boldsymbol{\rho}(t) = \boldsymbol{U}(0 \to t)\boldsymbol{\rho}_0 \boldsymbol{U}(0 \to t)^\dagger. \tag{61}
$$

We plot the evolution along with the corresponding energy-power uncertainty relation in Fig. 5. In this case, the uncertainty bound is saturated at all times. The effect of periodically measuring the energy is also shown in Fig. 5. (An explicit expression for the commutator term within the uncertainty bound as well as comments on the consequences of energy measurement are discussed in Appendix C.) In line with the discussion in Sec. V C, after every measurement in the basis of $\boldsymbol{H}_0$, $\sigma_{\boldsymbol{H}_0} = 0$, but with $\sigma_{\boldsymbol{P}_B} \neq 0$. Since each measurement results in a complete loss of coherence, the bound also vanishes after each measurement.

### 2. Spin-Boson Model

The spin-boson model describes a single spin interacting with an environment of harmonic oscillators, and serves as a canonical model to illustrate decoherence and dissipation in quantum systems [73, 74]. We consider an open quantum battery described by this model.

The time evolution of the spin's density matrix is determined by the Lindblad master equation:

$$
\begin{aligned}
\frac{\mathrm{d}\boldsymbol{\rho}_S(t)}{\mathrm{d}t} &= -\frac{i}{\hbar}[\boldsymbol{H}_S, \boldsymbol{\rho}_S(t)] \\
&\quad + \gamma \boldsymbol{\sigma}^z \boldsymbol{\rho}_S(t)(\boldsymbol{\sigma}^z)^\dagger - \gamma \boldsymbol{\rho}_S(t),
\end{aligned} \tag{62}
$$

where $\boldsymbol{H}_S$ is the system Hamiltonian. We take

$$
\begin{aligned}
\boldsymbol{H}_S &= \boldsymbol{H}_0 + \boldsymbol{V}_S(t). & \text{(63a)} \\
\text{with } \boldsymbol{H}_0 &= \alpha_3 \boldsymbol{\sigma}^z & \text{(63b)} \\
\text{and } \boldsymbol{V}_S(t) &= \alpha_1 \boldsymbol{\sigma}^x. & \text{(63c)}
\end{aligned}
$$

The density matrix $\boldsymbol{\rho}_S(t)$ takes the same form as (60) with a time-dependent $\vec{\beta}$.

Plugging into (62) yields the coupled differential equations:

$$
\dot{\vec{\beta}}(t) = \frac{1}{\hbar}\left(\vec{\alpha} \times \vec{\beta}(t)\right) - \vec{\gamma} \cdot \vec{\beta},
$$
$$
\text{where } \vec{\gamma} := \begin{pmatrix} \gamma & \gamma & 0 \end{pmatrix}^T. \tag{64}
$$

The form of these equations is the same as the Bloch equations [75] that arise in the context of nuclear magnetic resonance. Exact solutions can be obtained (see, e.g., Refs. [76, 77]) for all parameter ranges. It follows from the methods in Ref. [77] that all the solutions $\beta(t)$ of Eq. (64) converge at $t \to \infty$ to $\beta(\infty) = 0$. To see this, note that Eq. (64) can also we written as

$$
\begin{pmatrix} \dot{\beta}_1(t) \\ \dot{\beta}_2(t) \\ \dot{\beta}_3(t) \end{pmatrix} + \underbrace{\begin{pmatrix} \gamma & \frac{\alpha_3}{\hbar} & 0 \\ -\frac{\alpha_3}{\hbar} & \gamma & \frac{\alpha_1}{\hbar} \\ 0 & -\frac{\alpha_1}{\hbar} & 0 \end{pmatrix}}_{\Gamma} \begin{pmatrix} \beta_1(t) \\ \beta_2(t) \\ \beta_3(t) \end{pmatrix} = 0 \tag{65}
$$

The steady-state solution, obtained by setting $\dot{\vec{\beta}} = 0$ is readily seen to be $\vec{\beta} = 0$, regardless of the initial condition $\vec{\beta}(0)$. Thus, as $t \to \infty$, we have $\rho_S \to \mathbb{I}/2$. The exact functional form of the decay of $\vec{\beta}(t)$ towards 0, which can be under-damped, critically damped, or over-damped, depends on the parameter values $\gamma, \alpha_1$ and $\alpha_3$. (See for example, Ref. [77].)

To evaluate the energy-power uncertainty relationship, let us first obtain an expression for the power operator. The contribution to the power operator due to the closed system dynamics is

$$
\begin{aligned}
\boldsymbol{P}_B^c &= -\frac{i}{\hbar}[\boldsymbol{H}_0, \boldsymbol{V}_S] & \text{(66a)} \\
&= -\frac{i\alpha_3\alpha_1}{\hbar}[\boldsymbol{\sigma}^z, \boldsymbol{\sigma}^x] & \text{(66b)} \\
&= \frac{2\alpha_3\alpha_1}{\hbar}\boldsymbol{\sigma}^y. & \text{(66c)}
\end{aligned}
$$

The power contribution due to the dissipative terms is

$$
\begin{aligned}
\boldsymbol{P}_B^o &= \gamma \boldsymbol{L}^\dagger \boldsymbol{H}_S \boldsymbol{L} - \frac{\gamma}{2}\{\boldsymbol{H}_S, \boldsymbol{L}^\dagger \boldsymbol{L}\} & \text{(67a)} \\
&= \gamma \boldsymbol{\sigma}^z \boldsymbol{H}_S \boldsymbol{\sigma}^z - \gamma \boldsymbol{H}_S & \text{(67b)} \\
&= \gamma(-\alpha_1 \boldsymbol{\sigma}^x + \alpha_3 \boldsymbol{\sigma}^z) - \gamma(\alpha_1 \boldsymbol{\sigma}^x + \alpha_3 \boldsymbol{\sigma}^z) & \text{(67c)} \\
&= -2\alpha_1 \gamma \boldsymbol{\sigma}^x & \text{(67d)}
\end{aligned}
$$

wherein we used $(\boldsymbol{\sigma}^z)^3 = \boldsymbol{\sigma}^z$ and $\boldsymbol{\sigma}^z \boldsymbol{\sigma}^x \boldsymbol{\sigma}^z = -\boldsymbol{\sigma}^z$ in the third step. Thus, the total power operator is

$$
\begin{aligned}
\boldsymbol{P}_B &= \boldsymbol{P}_B^c + \boldsymbol{P}_B^o \\
&= -2\alpha_1 \gamma \boldsymbol{\sigma}^x + \frac{2\alpha_3\alpha_1}{\hbar}\boldsymbol{\sigma}^y,
\end{aligned} \tag{68}
$$

from which we obtain the energy-power commutator

$$
[\boldsymbol{H}_0, \boldsymbol{P}_B] = -4i\alpha_1\alpha_3\left(\gamma \boldsymbol{\sigma}^y + \frac{\alpha_3}{\hbar}\boldsymbol{\sigma}^x\right). \tag{69}
$$

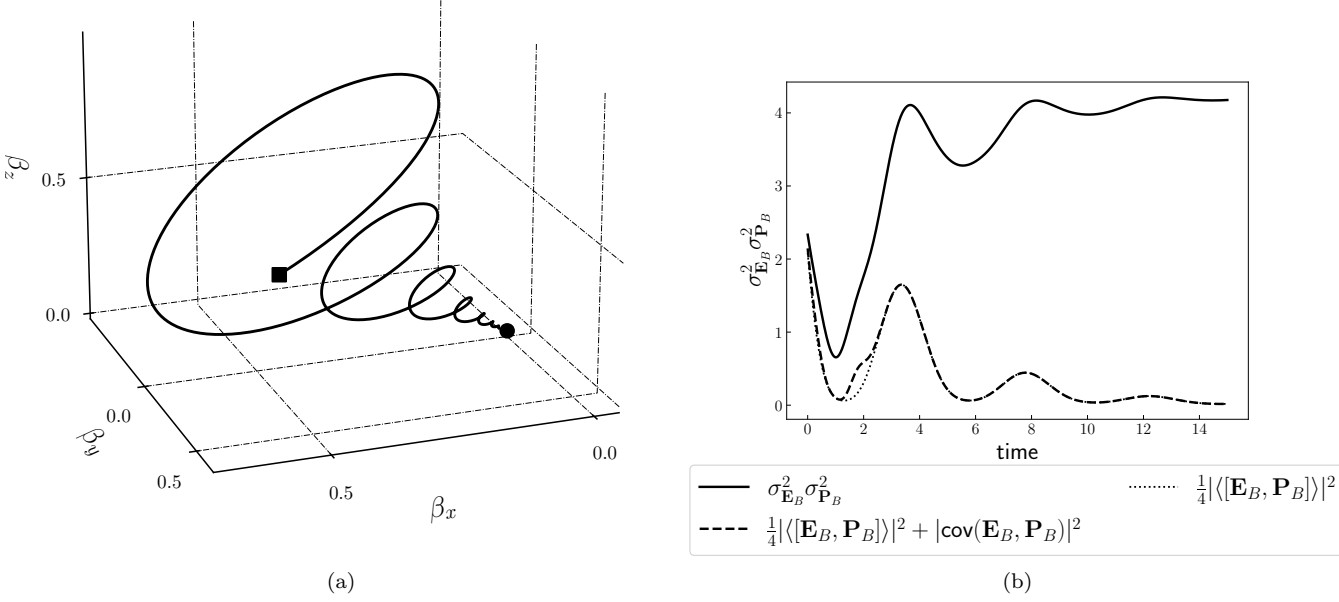

(a)

(b)

FIG. 6. (a) Numerically integrated solution trajectory $\beta(t)$ of the Bloch equations Eq. (64) for $\gamma = 0.25$ and $\alpha_1 = \alpha_3 = \hbar = 1$. The black square denotes the initial value $\vec{\beta}(0) = \frac{1}{\sqrt{3}}(1, 1, 1)^T$. The evolution (black curve) converges to the steady state, $\vec{\beta}(\infty) = 0$ (black circle). (b) A plot of the evolution of the product $\sigma^2_{\mathbf{E}_B} \sigma^2_{\mathbf{P}_B}$ as a function of time, along with the lower bound obtained from Eq. (48).

We now readily obtain the various quantities in the uncertainty relationship. First, we see that

$$\sigma^2_{\boldsymbol{E}_B} = \sigma^2_{\boldsymbol{H}_0} = \langle \boldsymbol{H}_0^2 \rangle - \langle \boldsymbol{H}_0 \rangle^2 \tag{70a}$$
$$= \alpha_3^2(1 - \beta_3(t)^2) \tag{70b}$$

The uncertainty in power evaluates to

$$\sigma^2_{\boldsymbol{P}_B} = \langle \boldsymbol{P}_B^2 \rangle - \langle \boldsymbol{P}_B \rangle^2 \tag{71a}$$
$$= (2\alpha_1)^2 \left[ \left( \gamma^2 - \frac{\alpha_3^2}{\hbar^2} \right) - \left( \gamma\beta_1(t) - \frac{\alpha_3\beta_2(t)}{\hbar} \right)^2 \right] \tag{71b}$$

On the other hand, we have

$$\frac{1}{4} |\langle [\boldsymbol{E}_B, \boldsymbol{P}_B] \rangle|^2 = (2\alpha_1\alpha_3)^2$$
$$\times \left( \frac{\alpha_3\beta_1(t)}{\hbar} + \gamma\beta_2(t) \right)^2, \tag{72a}$$

and $|\mathrm{cov}(\boldsymbol{E}_B, \boldsymbol{P}_B)|^2 = (2\alpha_1\alpha_3)^2 \beta_3(t)^2$
$$\times \left( \frac{\alpha_3\beta_2(t)}{\hbar} - \gamma\beta_1(t) \right)^2. \tag{72b}$$

The bound for the Robertson-Schrödinger uncertainty relation relation for $\boldsymbol{H}_0$ and $\boldsymbol{P}$, Eq. (48) is then obtained by summing up these two terms.

While it is not the purpose of this paper to delineate the various parameter regimes, we illustrate the damping

by numerically simulating the evolution for a fixed set of parameters. For concreteness, we choose $\gamma = 0.25$ and $\alpha_1 = \alpha_3 = \hbar$, so that the matrix $\Gamma$ becomes

$$\Gamma = \begin{pmatrix} 0.25 & 1 & 0 \\ -1 & 0.25 & 1 \\ 0 & -1 & 0 \end{pmatrix}. \tag{73}$$

The solution is then given by

$$\vec{\beta}(t) = \exp(-\Gamma t)\vec{\beta}(0). \tag{74a}$$
$$= S \begin{pmatrix} e^{-\lambda_1 t} & 0 & 0 \\ 0 & e^{-\lambda_2 t} & 0 \\ 0 & 0 & e^{-\lambda_3 t} \end{pmatrix} S^{-1} \vec{\beta}(0), \tag{74b}$$

where $\Gamma = S\Lambda S^{-1}$ is the diagonalization of $\Gamma$, with $\lambda_i$ being the eigenvalues of $\Gamma$. The eigenvalues are approximately $0.124$ and $0.188 \pm 1.407i$, all of which have positive real parts. Consequently, $e^{-\Lambda t} \to 0$ as $t \to \infty$, as expected. We plot the evolution $\vec{\beta}(t)$ for an example initial value of $\vec{\beta}(0) = \frac{1}{\sqrt{3}}(1, 1, 1)^T$ in Fig. 6a.

Let us now consider the various values in Eq. (48) in the limit of $t \to \infty$. Following the discussion in Sec. V C, expect the commutator term to approach 0, since the qubit loses all coherence and becomes a maximally mixed state. This is precisely what we get (along with the covariance term becoming zero) upon plugging in $\vec{\beta} = 0$ in (72). We also expect $\langle \boldsymbol{E}_B \rangle$ to equal zero since the spectrum of the battery Hamiltonian is symmetric about 0.

$\langle \boldsymbol{P}_B \rangle$ also equals zero since there cannot be any charging or discharging in the steady-state limit. However, the variances are not expected to be zero, since the second moment of both the operators is non-zero, even for the steady-state (i.e., $\vec{\beta} = 0$). Indeed, for the example considered here, we get $\sigma_E = 1$ and $\sigma_P = 2$ upon plugging in $\vec{\beta} = 0$ in (70b) and (71b) respectively. We plot the numerically obtained evolution of the uncertainty in Fig. 6b.

Finally, we focus on the heat flow, (thermodynamic) power and the entropy rate of Eq. (22d). Since $\boldsymbol{H}_S$ is time-independent, the power is $\mathring{\mathcal{W}} = 0$. (Consequently, the power uncertainty $\sigma_{\mathring{\mathcal{W}}} = 0$ as well.) On the other hand, the entropy rate and heat flow are in general non-zero, and do not commute with each other:

$$
\begin{aligned}
&[\mathcal{D}^*(\boldsymbol{H}_S), \mathcal{D}^*(\log \boldsymbol{\rho})] \\
&= \gamma^2 \Big( \boldsymbol{\sigma}^3 [\boldsymbol{H}_S, \log \boldsymbol{\rho}] \boldsymbol{\sigma}^3 + [\boldsymbol{H}_S, \log \boldsymbol{\rho}] \\
&\quad + [\boldsymbol{H}_S \boldsymbol{\sigma}^3, (\log \boldsymbol{\rho}) \boldsymbol{\sigma}^3] + [\boldsymbol{\sigma}^3 \boldsymbol{H}_S, \boldsymbol{\sigma}^3 (\log \boldsymbol{\rho})] \Big).
\end{aligned}
\tag{75}
$$

Using that $[\boldsymbol{\rho}, \log \boldsymbol{\rho}] = 0$, we have

$$
\mathcal{B} = \frac{\gamma^4}{4} |\operatorname{Tr}(\boldsymbol{\sigma}^3 [\boldsymbol{H}_S, \log \boldsymbol{\rho}] \boldsymbol{\sigma}^3 \boldsymbol{\rho})|^2.
\tag{76}
$$

If $\boldsymbol{V} = 0$, the expression above reduces to a simpler expression

$$
\begin{aligned}
&[\mathcal{D}^*(\boldsymbol{H}_S), \mathcal{D}^*(\log \boldsymbol{\rho})] \\
&= \gamma^2 \alpha_3 \Big( \boldsymbol{\sigma}^3 [\boldsymbol{\sigma}^3, \log \boldsymbol{\rho}] \boldsymbol{\sigma}^3 + [\boldsymbol{\sigma}^3, \log \boldsymbol{\rho}] \Big)
\end{aligned}
\tag{77}
$$

and thus the commutator term in the uncertainty relation for entropy rate and heat flow is

$$
\mathcal{B} = \frac{\gamma^4 \alpha_3^2}{4} |\operatorname{Tr}(\boldsymbol{\sigma}^3 [\boldsymbol{\sigma}^3, \log \boldsymbol{\rho}] \boldsymbol{\sigma}^3 \boldsymbol{\rho})|^2
\tag{78}
$$

While the entropy rate is not a quantum mechanical observable in the sense of being a state-independent Hermitian operator, $\mathcal{B}$ still lower bounds the product of uncertainties of entropy rate and heat flow, since the Robertson-Schrödinger uncertainty relation is applicable to any pair of Hermitian operators.

## VI. TYPICAL UNCERTAINTY VALUES

The power-energy uncertainty relationships in Eq. (48) that constrain quantum batteries, and the power, heat flow, and internal energy uncertainty relations in Eqs. (30) and (27) depend on the typically time-dependant state. This makes evaluating these bounds hard, since it requires full knowledge of the generally-complex time evolution of the system. To sidestep this challenge, we evaluate the typical values of the uncertainty relations. Specifically, we employ Weingarten calculus, which allows us to integrate over the unitary group

effectively. This enables us to average out the complex behavior arising from the time evolution and focus on the statistical properties of the quantum states and operations, and look at a simpler uncertainty probe measure. Since we are interested in the quantum origin of the uncertainty, we focus only on the generalized Heisenberg uncertainty term.

First, it is easy to see that given a (scalar) random variable $a$, if $a \geq 0$, then $\overline{a} \geq 0$, so that

$$
\overline{\sigma_{\boldsymbol{A}}^2 \sigma_{\boldsymbol{B}}^2} \geq \overline{\left| \frac{1}{2i} \langle [\boldsymbol{A}, \boldsymbol{B}] \rangle \right|^2}
\tag{79}
$$

Thus, the average uncertainty $\sigma_{\boldsymbol{A}}^2 \sigma_{\boldsymbol{B}}^2$ relationship is lower bounded by the uncertainty *probe*

$$
\overline{\mathcal{B}}_* = \overline{\left| \frac{1}{2i} \langle [\boldsymbol{A}, \boldsymbol{B}] \rangle \right|^2}
\tag{80}
$$

which is the quantity we focus on in this section. The star at the bottom will denote the operator in which randomness is introduced. The average above, as we show below, is intended as the average over the unitary channel, entering either via the initial condition $\boldsymbol{\rho}_0$ or via the interactions $\boldsymbol{V}_S$ and $\boldsymbol{V}_{SE}$. Depending on how the averaging is performed, we will use a different suffix of the uncertainty probe.

### A. Isospectral Twirling

Before discussing the average values, we introduce the techniques we employ for the isospectral twirling of the uncertainty probe of Eq. (79). Let us briefly describe the Haar measure averages we perform in the following. We will use the uncertainty probe as a measure of the average uncertainty given an initial condition with a given purity, in terms of the $2k$-isospectral twirling of a unitary channel is the average of $(U^\dagger \boldsymbol{G} U)^{\otimes k}$ over $U$ sampled uniformly from the unitary group.

Let us define this average formally. Let $\mathcal{H} \simeq \mathbb{C}^d$ be a $d$-dimensional Hilbert space and let $\boldsymbol{G} \in \mathcal{H}$ be an operator. The $k$-isospectral twirling of $\boldsymbol{G}$ is defined as

$$
\begin{aligned}
\hat{\mathcal{R}}^{(k)}(\boldsymbol{G}) &= \int \mathrm{d}U \, U^{\dagger \otimes k} \left( \boldsymbol{G}^k \right) U^{\otimes k} \\
&\equiv \overline{(U^\dagger)^{\otimes k} \boldsymbol{G}^{\otimes k} U^{\otimes k}}^U
\end{aligned}
\tag{81}
$$

and $\mathrm{d}U$ represents the Haar measure over the unitary group $\mathcal{U}(d)$, where the overline represents a shorthand notation for these integrals. We denote the operator being averaged without the bold notation in the following for clarity.

The quantity $\hat{\mathcal{R}}^{(k)}(\boldsymbol{G})$ is the isospectral twirling of $\boldsymbol{G}$, that is, the average over its $k$-fold channel [78, 79]. We know from Weingarten calculus that

$$
\hat{\mathcal{R}}^{(k)}(\boldsymbol{G}) := \sum_{\pi\sigma \in \mathcal{S}_k} W_g^U(\pi\sigma^{-1}, d) \operatorname{tr}\left( T_\pi^{(k)} \boldsymbol{G}^{\otimes k} \right) T_\sigma^{(k)}
\tag{82}
$$

where the sum over $\pi, \sigma$ of the permutation group $\mathcal{S}_k$, and $W_g^U(\sigma', d)$ is the Weingarten coefficient associated with the permutation $\sigma'$ (in the cycle representation, see [79, 80] for details). The key formulae used in the following are derived in Appendix D.

The average uncertainty $\sigma_A^2 \sigma_B^2$ relationship is lower bounded by the uncertainty *probe*. In order to obtain a 'typical' value of the bound, we average over the initial condition by averaging over the random initial state, $\boldsymbol{\rho}_0^U = U^\dagger \boldsymbol{\rho}_0 U$. We thus have

$$\overline{\mathcal{B}}_\rho = \overline{\left| \frac{1}{2i} \langle [\boldsymbol{A}, \boldsymbol{B}] \rangle \right|^2} \equiv \overline{\left| \frac{1}{2i} \operatorname{tr}\{ [\boldsymbol{A}, \boldsymbol{B}] U^\dagger \boldsymbol{\rho}_0 U \} \right|^2}^U \quad (83a)$$

$$= \frac{1}{4} \operatorname{tr}\left( [\boldsymbol{A}, \boldsymbol{B}] \otimes [\boldsymbol{B}^\dagger, \boldsymbol{A}] \hat{\mathcal{R}}^2(\boldsymbol{\rho}_0) \right), \quad (83b)$$

which is one of the probes we focus on in the following.

To calculate the average uncertainty probe, we employ the technique of isospectral twirling. This involves averaging over the unitary group with respect to the Haar measure, as described previously. Specifically, for a given operator $\boldsymbol{G}$, the isospectral twirling operation $\hat{\mathcal{R}}^{(k)}(\boldsymbol{G})$ provides a way to average over all possible unitary transformations while preserving the spectrum of $\boldsymbol{G}$. By applying this operation, we can effectively average the uncertainty probe $\overline{\mathcal{B}}_*$ over the unitary group, ensuring that the resulting bound is independent of the specific time evolution details of the system.

This approach simplifies the quantification of the uncertainty bounds, as it removes the need for complete knowledge of the system's time evolution. Instead, we rely on the properties of the Haar measure and the permutation operators to derive general results that hold for any unitary channel with a given spectrum. This makes the isospectral twirling technique a powerful tool for analyzing the average uncertainty in quantum systems, particularly in cases where the exact dynamics are complex or unknown.

### B. Closed system case

We first assume a random initial state as a twirl of the form

$$\boldsymbol{\rho}_0^U = U \boldsymbol{\rho}_0 U^\dagger, \quad (84)$$

and because of the invariance of the measure, it can be immediately seen that time evolution must be absent in the final result. We show in Appendix D that

$$\overline{\mathcal{B}}_\rho = \frac{l_s X}{(2\hbar)^2} \quad (85)$$

where

$$X = \operatorname{Tr}\left( \boldsymbol{H}_0^2 \left( 6 \boldsymbol{V}_S \boldsymbol{H}_0^2 \boldsymbol{V}_S - 8(\boldsymbol{H}_0 \boldsymbol{V}_S)^2 + 2 \boldsymbol{H}_0^2 \boldsymbol{V}_S^2 \right) \right)$$

and where

$$l_s = \frac{d_S \mathcal{P} - 1}{d_S(d_S^2 - 1)}. \quad (86)$$

with

$$\mathcal{P} = \operatorname{Tr} \boldsymbol{\rho}_0^2 \quad (87)$$

being the initial state purity. It is easy to see that if our initial state is fully mixed, i.e., $\mathcal{P} = \frac{1}{d_S}$, then $\mathcal{B} = 0$. In the limit $d_S \gg 1$ (and for $\mathcal{P}$ finite), the expression above reduces to

$$\overline{\mathcal{B}}_\rho \approx \frac{\mathcal{P} X}{(2 d_S \hbar)^2} \quad (88)$$

For instance, for the case of the single qubit of Sec. V D 1, with $\boldsymbol{H_0} = \alpha_3 \boldsymbol{\sigma}^z$ and $\boldsymbol{V} = v_1 \sigma^1$

$$X = 32 \alpha_3^4 v_1^2 \quad (89)$$

and thus

$$\overline{\mathcal{B}}_\rho = \frac{4(2\mathcal{P} - 1)}{3\hbar^2} \alpha_3^3 v_1^2. \quad (90)$$

Thus, if the initial state is completely mixed, the right-hand side of the uncertainty relationship is zero. If we look at the exact calculation instead, it is easy to see that for $\vec{\beta} = 0$, then $\boldsymbol{\rho}_t = \frac{\mathbb{I}}{2}$. Since $[\boldsymbol{H}_0, [\boldsymbol{H}_0, \boldsymbol{V}_S]] = 4\alpha_3^2 v_1 \boldsymbol{\sigma}^1$, we have $\langle [\boldsymbol{H}_0, [\boldsymbol{H}_0, \boldsymbol{V}_S]] \rangle_t = 0$, which is consistent with the calculation above. When instead the unitary channel is applied to the interaction $\boldsymbol{V}_S$, the probe $\overline{\mathcal{B}}_V$ remains time-dependent, as we show in Appendix D. This makes this probe as complicated as the full dynamics to analyze.

### C. Open system case

Let us now consider this probe in the case of the open system, as in Sec. II A. Our assumption is that the density matrix is of the from $\boldsymbol{\rho} = \boldsymbol{\rho}_S \otimes \boldsymbol{\rho}_E$, e.g. that there is no entanglement between the subsystem and the environment, and the interaction potential is of the form $\boldsymbol{V}_{SE} = \boldsymbol{V}_S \otimes \boldsymbol{V}_E$. In this setup, it is easy to see that, if the unitary channel acts on the initial state of the subsystem, we have as shown in Appendix D that the uncertainty probe is given by:

$$\overline{\mathcal{B}}_\rho = \frac{l_s}{(2\hbar)^2} \operatorname{Tr}\left( [\boldsymbol{H}_0, [\boldsymbol{H}_0, \boldsymbol{V}_S]]^2 \right) \cdot | \operatorname{Tr}\left( \boldsymbol{\rho}_E \boldsymbol{V}_E \right) |^2. \quad (91)$$

where $l_s$ is defined analogously to before. Thus, in this setup, we see that the probe naturally factories in the product of two terms.

If instead, the unitary channel operates on the interaction potential $U^\dagger \boldsymbol{V}_E U$. Then we have

$$\overline{\mathcal{B}}_V = \frac{\overline{\mathcal{D}}_E}{4} | \operatorname{Tr}\left( [\boldsymbol{H_0}, [\boldsymbol{H_0}, \boldsymbol{V}_S]] \boldsymbol{\rho}_S(t) \right) |^2 \quad (92)$$

where, if the dimension of the bath satisfies $d_E \gg 1$

$$\overline{\mathcal{D}}_E = \frac{\text{Tr}(\boldsymbol{V}_E^2)(1 + \mathcal{P}_E) - d_E \Delta V_\infty^2}{d_E^2} \qquad (93)$$

and

$$\Delta V_\infty^2 = \frac{1}{d_E}\Big( \text{Tr}(\boldsymbol{V}_E^2) - \text{Tr}(\boldsymbol{V}_E)^2 \Big). \qquad (94)$$

Above, $\mathcal{P}_E = \text{Tr}(\boldsymbol{\rho}_E^2)$ is the purity of the bath. We see then that the probe of the uncertainty relationship is exactly the one of a closed system's quantum battery but with an overall factor that depends on the state of the bath. If the bath is in a thermal state, e.g. $\boldsymbol{\rho}_E = \frac{e^{-\beta \boldsymbol{H}_B}}{Z}$, the purity is a function of the inverse temperature $\beta$. If $\beta = 0$, $\mathcal{P}_E = 1/d_E$, while it is 1 at $\beta = \infty$.

## VII. CONCLUSIONS

Thermodynamic uncertainty relations (TURs) are a set of fundamental relations in stochastic thermodynamics that relate the fluctuations of thermodynamic currents to a system's entropy production for non-equilibrium steady-states. In this paper, we introduce a different class of TURs, which we call operator-based TURs. These TURs place lower bounds on the uncertainty of pairs of thermodynamic currents for any closed or open quantum system, at any point during the evolution of the systems (i.e., it does not assume non-equilibrium steady states).

To derive the operator-based TURs, we define Hermitian operators $\{\mathring{\mathcal{W}}, \mathring{\mathcal{Q}}, \mathring{\mathcal{U}}\}$ whose averages yield the work rate (i.e., power), heat flow and internal energy rate. The existence of these operators contrasts with the difficulty in defining observables that characterize work and heat exchanges over a finite time period. This distinction closely follows the fact that while work and heat are process-dependent, their rates are state functions. The operator-based TURs follow from the Robertson-Schrödinger uncertainty relation, which relates the variances of any two quantum mechanical observables to their commutator and covariance. These TURs thus capture fluctuations that are uniquely quantum meachanical, since non-commutativity, while inherent in quantum physics, is absent in classical physics.

For various simple models, the power and heat rate operators assume simple Hermitian forms that might be accessible experimentally. Furthermore, for these examples, we found uncertainty bounds for power-heat flow TURs that depend on the average system-environment interaction Hamiltonian, satisfying $\sigma_{\mathring{\mathcal{Q}}} \sigma_{\mathring{\mathcal{W}}} \geq a|\langle \boldsymbol{V}_{SE} \rangle|$ for $a \in \mathbb{R}^+$. This sort of quantum uncertainty relation is somewhat reminiscent of conventional TURs of the form Eq. (1).

To further illustrate the utility of the operator approach, we computed energy-power uncertainty relations for open and closed quantum batteries. Similar to the position-momentum uncertainty relation that underlies the Heisenberg uncertainty principle, we find the counter-intuitive result that a lower uncertainty in the battery energy is accompanied by a larger uncertainty in battery charging power, even though power is related to the time derivative of battery energy.

Next, we commented on the role of decoherence in the magnitude of the various bounds we found. Specifically, we showed that decoherence in the eigenbasis of either of the two operators in our operator-based TURs results in the bound becoming smaller, since the expectation value of the commutator of the two operators decreases with larger decoherence, becoming zero when the state of the system becomes fully decohered. In order to obtain an estimate for the typical values of the uncertainty bounds, we used Weingarten calculus to derive Haar-averaged bounds. Since our bounds require knowledge of the state of the system (i.e., the density matrix), such typical values can serve as useful guesses when the form of the density matrix cannot be solved exactly.

We leveraged the operator-based uncertainty relations to derive novel constraints on thermodynamics processes that apply beyond non-equilibrium steady states. The most suggestive future research direction involves bridging the gap between quantum operator-TURs and conventional TURs by understanding scenarios where one reduces to the other. It would be interesting to explore higher moments of these quantities and derive various fluctuation relations from an operatorial perspective for current operators. Another interesting direction would be to extend the definitions of power and heat flow to systems with infinite-dimensional Hilbert spaces, since the cyclic property of the trace, which we used extensively in this work, may not always be applicable to such systems. Given the rapid development of quantum computing devices, it would be interesting to verify and probe these TURs experimentally on gate-based quantum computers and quantum annealing devices.

## ACKNOWLEDGMENTS

We thank Tanmoy Biswas for helpful discussions. The authors benefited from the stimulating atmosphere of the Quantum to Cosmos workshop, organized at LANL. P.S. and F.C. acknowledge the support of NNSA for the U.S. DoE at LANL under Contract No. DE-AC52-06NA25396, and Laboratory Directed Research and Development (LDRD) for support through 20240032DR. L.P.G.P. acknowledges support by the LDRD program of LANL under project number 20230049DR, and Beyond Moore's Law project of the Advanced Simulation and Computing Program at LANL managed by Triad National Security, LLC, for the National Nuclear Security Administration of the U.S. DOE under contract 89233218CNA000001.

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

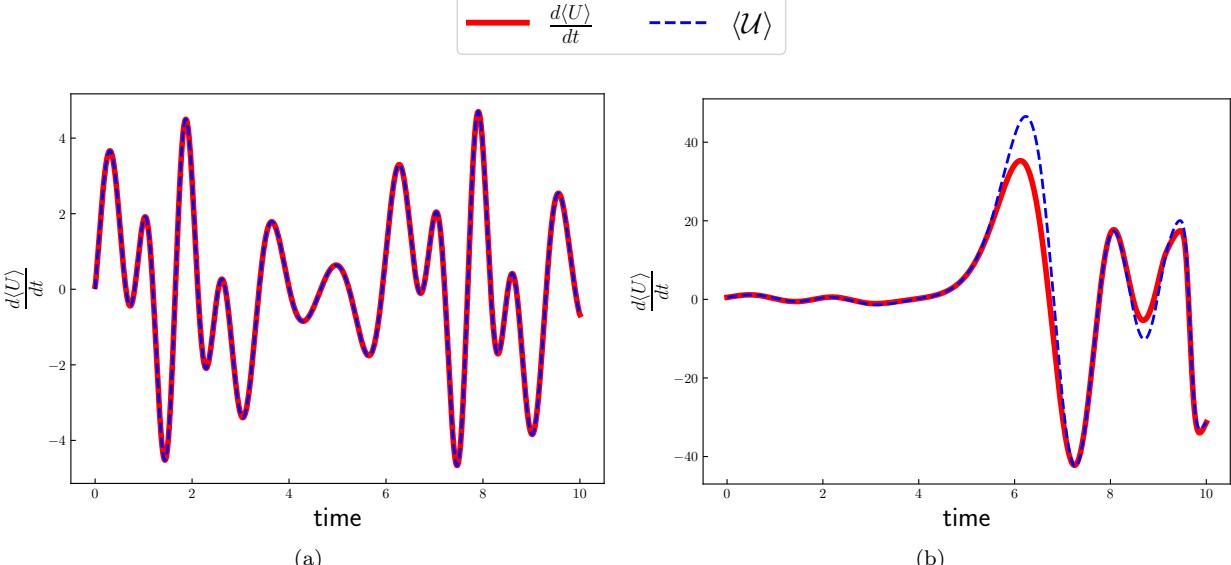

FIG. 7. Numerical verification of Eq. (A1) for (a) two interacting spins [Eq. (23)] and (b) two interacting harmonic oscillators [Eq. (25)]. For (a), we set $g = \hbar = 1$ and $f(t) = \sin t + 2$, and use the initial condition $|\psi(0)\rangle = |\uparrow\rangle \otimes |\uparrow\rangle$, with $|\uparrow\rangle$ denoting the eigenvalue 1 eigenvector of $\boldsymbol{\sigma}^z$. For (b) we set $\omega_a(t) = \sin(t) + 2$, with $\hbar = g = m = \omega_b = 1$. The initial condition at $t = 0$ is chosen to be $|\psi(0)\rangle = |0\rangle \otimes |0\rangle$, i.e. the tensor product of the ground states of the two oscillators. The discrepancy between the two curves in (b) arises due to numerical errors associated with truncation of the Hilbert space to low-energy subspaces of the two oscillators.

## Appendix A: Numerical Simulations for examples of interacting spins and oscillators

In this appendix, we present details of numerical simulations of the two examples (interacting spins and interacting oscillators) discussed in Sec. III C.

Additionally, for each example, we numerically show below that the operator corresponding to the rate of change of internal energy has an expectation value equal to the time derivative of the internal energy expectation value. Specifically, Eq. (5) in conjunction with Eq. (17) implies that

$$\frac{\mathrm{d}\langle \boldsymbol{U} \rangle}{\mathrm{d}t} = \langle \mathring{\boldsymbol{\mathcal{U}}} \rangle, \tag{A1}$$

with $\boldsymbol{U} = \boldsymbol{H}_S$ denoting the internal energy of the system. We now discuss the numerical simulations in some detail.

### Two interacting spins

Let us revisit the example of two interacting spins with a driving term applied to one of the spins, with a Hamiltonian given by Eq. (23). We find numerically that Eq. (A1) is satisfied for this example, as seen in Fig. 7a. Using numerical ODE integration, we first obtain the solution of the Schrodinger equation over a grid of 1000 equally spaced time instances between $t = 0$ and $t = 10$. Next, we compute $\langle \boldsymbol{U} \rangle$ and its derivative $\frac{\mathrm{d}\langle \boldsymbol{U} \rangle}{\mathrm{d}t}$ using a standard second-order approximation, at each point of the time grid. On the other hand, $\langle \mathring{\boldsymbol{\mathcal{U}}} \rangle$ was computed on the same time grid by computing the expectation value of $\mathring{\boldsymbol{\mathcal{U}}}$ from Eq. (24c).

As seen in Fig. 7a, there is a close match between the time derivative of average internal energy, and the expectation value of the energy rate operator. We note that a similar direct verification of the validity of our expressions for $\mathring{\boldsymbol{\mathcal{W}}}$ and $\mathring{\boldsymbol{\mathcal{Q}}}$ cannot be done, since as discussed in the introduction, the work done and heat exchanged over a finite period of time are not state functions, and consequently, they are not described by quantum mechanical observables.

## Two interacting oscillators

The Hamiltonian Eq. (25) describing two coupled, driven harmonic oscillators was expressed in terms of the position and momenta operators corresponding to the two oscillators. For numerical simulations, we will find it convenient to work with creation and annihilation operators instead, with the system and environment annihilation operators denoted by $\boldsymbol{a}$ and $\boldsymbol{b}$ respectively. The annihilation ladder operators for the two oscillators can be expressed as [81]

$$\boldsymbol{a}(t) = \sqrt{\frac{m\omega_a(t)}{2\hbar}}\left(\boldsymbol{x}_a + \frac{i}{m\omega_a(t)}\boldsymbol{p}_a\right), \tag{A2a}$$

$$\text{and } \boldsymbol{b} = \sqrt{\frac{m\omega_b}{2\hbar}}\left(\boldsymbol{x}_b + \frac{i}{m\omega_b}\boldsymbol{p}_b\right). \tag{A2b}$$

The operators $\boldsymbol{a}$ and $\boldsymbol{a}^\dagger$ are explicitly time-dependent due to the $\omega_a(t)$ terms. Henceforth, we will drop the time dependence of $\boldsymbol{a}$ and $\omega_a$ for notational simplicity.

In terms of the ladder operators, the Hamiltonian takes the form

$$\boldsymbol{H}_{\text{tot}} = \boldsymbol{H}_S \otimes \mathbb{I} + \boldsymbol{V}_{SE} + \mathbb{I} \otimes \boldsymbol{H}_E \tag{A3a}$$

$$\boldsymbol{H}_S = \hbar\omega_a\left(\boldsymbol{a}^\dagger\boldsymbol{a} + \frac{1}{2}\right), \tag{A3b}$$

$$\boldsymbol{H}_E = \hbar\omega_b\left(\boldsymbol{b}^\dagger\boldsymbol{b} + \frac{1}{2}\right), \tag{A3c}$$

$$\boldsymbol{V}_{SE} = \frac{\hbar g_x}{m\sqrt{\omega_a\omega_b}}(\boldsymbol{a} + \boldsymbol{a}^\dagger)(\boldsymbol{b} + \boldsymbol{b}^\dagger). \tag{A3d}$$

[As expected, the only term above that is time dependent above is $\boldsymbol{H}_S$. Even though $\boldsymbol{V}_{SE}$ appears to be time-dependent, the explicit time-dependence of $\boldsymbol{a}$ and $\boldsymbol{a}^\dagger$ from Eq. (A2a) cancels with the time dependence of the $\omega_a$ term in Eq. (A3d).]

The power and heat-flux operators [see Eq. (26)] can thus be equivalently expressed in terms of the ladder operators as

$$\mathring{\boldsymbol{\mathcal{W}}} = \frac{\hbar\dot{\omega}_a}{2m}(\boldsymbol{a}^\dagger + \boldsymbol{a})^2, \tag{A4a}$$

$$\text{and } \mathring{\boldsymbol{\mathcal{Q}}} = -i\hbar\frac{g}{m}\sqrt{\frac{\omega_a}{\omega_b}}(\boldsymbol{a}^\dagger - \boldsymbol{a})(\boldsymbol{b}^\dagger + \boldsymbol{b}). \tag{A4b}$$

We note that the Hamiltonian dynamics corresponding to Eq. (A3) can be solved exactly using the techniques developed in Ref. [66]. However, we restrict our analysis to a numerical study. Similar to the previous example, we numerically simulate the evolution of two interacting oscillators and compare the obtained values of $\frac{d\langle \boldsymbol{U}\rangle}{dt}$ with those of $\langle\mathring{\boldsymbol{\mathcal{U}}}\rangle$ (see Fig. 7b). For this simulation, we considered a truncated Hilbert space consisting of the lowest 100 energy levels of each of the two oscillators (so that the total Hilbert space dimension used was 100000), and solved the Schrödinger equation with Hamiltonian Eq. (A3) truncated to these levels. Consequently, while the two curves match, due to this approximation, we find that at later times, $\langle\mathring{\boldsymbol{\mathcal{U}}}\rangle$ deviates from the derivative of mean internal energy. Another potential source of error could be the fact that the operator definitions $\mathring{\boldsymbol{\mathcal{W}}}$, $\mathring{\boldsymbol{\mathcal{Q}}}$ and $\mathring{\boldsymbol{\mathcal{U}}}$ were based on the cyclic property of matrix traces, which does not always hold for infinite dimensional matrices. However, for any finite but sufficiently large truncation of Eq. (A3), we expect to Eq. (A1) to be satisfied upto numerical errors.

## Appendix B: Bounds on $\sigma_{\mathring{\mathcal{U}}}$ in terms of $\sigma_{\mathring{\mathcal{W}}}$ and $\sigma_{\mathring{\mathcal{Q}}}$

Here, we will obtain an upper bound on $\sigma_{\mathring{\mathcal{U}}}^2$ in terms of $\sigma_{\mathring{\mathcal{W}}}$ and $\sigma_{\mathring{\mathcal{Q}}}$. First, we note that

$$\mathring{\boldsymbol{\mathcal{U}}} = \mathring{\boldsymbol{\mathcal{W}}} + \mathring{\boldsymbol{\mathcal{Q}}}, \tag{B1a}$$

$$\text{and } \langle\mathring{\boldsymbol{\mathcal{U}}}\rangle = \langle\mathring{\boldsymbol{\mathcal{W}}}\rangle + \langle\mathring{\boldsymbol{\mathcal{Q}}}\rangle. \tag{B1b}$$

Taking the square of Eq. (B1a) and subtracting from it the square of Eq. (B1b), we get

$$\langle \mathring{\boldsymbol{\mathcal{U}}}^2 \rangle - \langle \mathring{\boldsymbol{\mathcal{U}}} \rangle^2 = \langle \mathring{\boldsymbol{\mathcal{W}}}^2 \rangle - \langle \mathring{\boldsymbol{\mathcal{W}}} \rangle^2 + \langle \mathring{\boldsymbol{\mathcal{Q}}}^2 \rangle - \langle \mathring{\boldsymbol{\mathcal{Q}}} \rangle^2 \tag{B2}$$
$$\langle \{ \mathring{\boldsymbol{\mathcal{Q}}}, \mathring{\boldsymbol{\mathcal{W}}} \} \rangle - 2 \langle \mathring{\boldsymbol{\mathcal{Q}}} \rangle \langle \mathring{\boldsymbol{\mathcal{W}}} \rangle$$
$$\text{that is } \sigma_{\mathring{\boldsymbol{\mathcal{U}}}}^2 = \sigma_{\mathring{\boldsymbol{\mathcal{W}}}}^2 + \sigma_{\mathring{\boldsymbol{\mathcal{Q}}}}^2 + 2\mathrm{cov}(\mathring{\boldsymbol{\mathcal{Q}}}, \mathring{\boldsymbol{\mathcal{W}}}). \tag{B3}$$

Adding and subtracting $2\sigma_{\mathring{\boldsymbol{\mathcal{Q}}}}\sigma_{\mathring{\boldsymbol{\mathcal{W}}}}$, we have

$$\sigma_{\mathring{\boldsymbol{\mathcal{U}}}}^2 = (\sigma_{\mathring{\boldsymbol{\mathcal{Q}}}} + \sigma_{\mathring{\boldsymbol{\mathcal{W}}}})^2 - 2(\sigma_{\mathring{\boldsymbol{\mathcal{Q}}}}\sigma_{\mathring{\boldsymbol{\mathcal{W}}}} - \mathrm{cov}(\mathring{\boldsymbol{\mathcal{Q}}}, \mathring{\boldsymbol{\mathcal{W}}})) \tag{B4}$$

or alternatively,

$$\sigma_{\mathring{\boldsymbol{\mathcal{U}}}}^2 = (\sigma_{\mathring{\boldsymbol{\mathcal{Q}}}} - \sigma_{\mathring{\boldsymbol{\mathcal{W}}}})^2 + 2(\sigma_{\mathring{\boldsymbol{\mathcal{Q}}}}\sigma_{\mathring{\boldsymbol{\mathcal{W}}}} + \mathrm{cov}(\mathring{\boldsymbol{\mathcal{Q}}}, \mathring{\boldsymbol{\mathcal{W}}})) \tag{B5}$$

From this, we obtain

$$\sigma_{\mathring{\boldsymbol{\mathcal{W}}}}\sigma_{\mathring{\boldsymbol{\mathcal{Q}}}} \pm \mathrm{cov}(\mathring{\boldsymbol{\mathcal{Q}}}, \mathring{\boldsymbol{\mathcal{W}}}) \geq t_{\pm} \geq 0, \tag{B6}$$

Eq. (B4) and (B5) thus give us

$$(\sigma_{\mathring{\boldsymbol{\mathcal{Q}}}} - \sigma_{\mathring{\boldsymbol{\mathcal{W}}}})^2 + 2t_+ \leq \sigma_{\mathring{\boldsymbol{\mathcal{U}}}}^2 \leq (\sigma_{\mathring{\boldsymbol{\mathcal{Q}}}} + \sigma_{\mathring{\boldsymbol{\mathcal{W}}}})^2 - 2t_- \tag{B7}$$

## Appendix C: Exact expressions for the single qubit quantum battery example

In this section, we obtain an exact expression for the commutator term for the energy-power uncertainty relation for the example of a single qubit quantum battery with the Hamiltonian Eq. (59). Let us first provide some further background on the expression in Eq. (38) of the main text for generic quantum batteries. If we write the matrices $\boldsymbol{V}_S(t)$ and $\boldsymbol{\rho}(t)$ in the basis of the battery Hamiltonian $\boldsymbol{H}_0$, retaining only the commutator term in Eq. (38), we have

$$\sigma_{\boldsymbol{E}_B}\sigma_{\boldsymbol{P}_B^c} \geq \frac{|\sum_{i \neq j}(\epsilon_i - \epsilon_j)^2 V_{ij}\rho_{ji}|}{2\hbar} \tag{C1a}$$

$$= \frac{\left|\sum_{i<j}(\epsilon_i - \epsilon_j)^2 \mathrm{Re}(V_{ij}\rho_{ji})\right|}{\hbar}, \tag{C1b}$$

where $\epsilon_i$s denote the eigenvalues of $\boldsymbol{H}_0$, and $V_{ij}$ and $\rho_{ij}$ denote the matrix elements of $\boldsymbol{V}_S(t)$ and $\boldsymbol{\rho}(t)$ respectively. The uncertainty bound clearly depends on the amount of decoherence in the $\boldsymbol{H}_0$ basis. We note that measurements in the $\boldsymbol{H}_0$ basis make the uncertainty of the energy $\sigma_{\boldsymbol{E}_B}^2 = 0$ just after the measurement. However, since the battery power operator is not diagonal in this basis in general, we can still have $\sigma_{P_B}^2 \geq 0$, as seen in Fig. 5. The right-hand side of Eq. (C1) is also seen to vanish as expected.

Assuming that at time $t = 0$, the state of the qubit is given by $\boldsymbol{\rho}_0 = \frac{1}{2}(\mathbb{I} + \vec{\sigma}.\vec{\beta})$, using Eq. (C1), we find that the commutator term in the uncertainty bound at time $t$ is given by

$$\begin{aligned}
\frac{|\sum_{i \neq j}(\epsilon_i - \epsilon_j)^2 \rho_{ij}v_{ji}|}{2} &= 4h_3^2 \mathrm{Re}\{V_{12}\rho_{21}(t)\} \\
&= \frac{2h_3^2}{\alpha^2} \times \big[ h_3^2(\beta_1 v_1 + v_1 - \beta_2 v_2) + (3\beta_1 + 1)v_1^3 \\
&\quad + (3\beta_1 + 1)v_1 v_2^2 + (\beta_1 + 1)v_1 v_3^2 \\
&\quad + 2h_3 \left( \beta_3 \left( v_1^2 + v_2^2 \right) + v_3(\beta_1 v_1 + v_1 - \beta_2 v_2) \right) \\
&\quad + v_1^2(\beta_2 v_2 + 2\beta_3 v_3) + v_2 \left( \beta_2 v_2^2 + 2\beta_3 v_2 v_3 - \beta_2 v_3^2 \right) \\
&\quad + \cos(2t\alpha)t_1 + \sin(2t\alpha)t_2 \big],
\end{aligned} \tag{C2}$$

where

$$t_1 = h_3^2(3\beta_1 v_1 + v_1 + \beta_2 v_2) +$$
$$(\beta_1 + 1)v_1^3 + (\beta_1 + 1)v_1 v_2^2 + (3\beta_1 + 1)v_1 v_3^2$$
$$- 2\beta_3 h_3 \left(v_1^2 + v_2^2\right) + 2h_3 v_3(3\beta_1 v_1 + v_1 + \beta_2 v_2)$$
$$- \left(v_1^2(\beta_2 v_2 + 2\beta_3 v_3)\right)$$
$$+ v_2 \left(-\beta_2 v_2^2 - 2\beta_3 v_2 v_3 + \beta_2 v_3^2\right)$$
$$\text{and } t_2 = \alpha \left(2\beta_3 v_1(h_3 + v_3) + (2\beta_1 + 1)\left(v_1^2 + v_2^2\right)\right) \tag{C3}$$

## Appendix D: Haar averages

### Key formulae

Let us introduce the key formulae for the Haar average used in the main text. First, we will use $\overline{X}$ to indicate the expectation value over the Unitary Haar measure of a certain quantity, where $U$ is the unitary operator being averaged. In particular, we use the following formulae [79, 82]:

$$\overline{\mathbf{1}} = \mathbf{1} \tag{D1a}$$

$$\hat{\mathcal{R}}^1(U) = \overline{U A U^\dagger} = \text{Tr}(A)\mathbb{I} \tag{D1b}$$

$$\overline{(U \otimes U)(A \otimes B)(U^\dagger \otimes U^\dagger)} = \lambda_+(A \otimes B)\mathbf{\Pi}_+ + \lambda_-(A \otimes B)\mathbf{\Pi}_- \tag{D1c}$$

where

$$\mathbf{\Pi}_\pm = \frac{\mathbb{I}^{\otimes 2} \pm \mathbb{S}}{2} \tag{D2}$$

where $\mathbb{S}$ is the swap operator, and where

$$\lambda_\pm(A \otimes B) = \frac{\text{Tr}((A \otimes B)\mathbf{\Pi}_\pm)}{\text{Tr}(\mathbf{\Pi}_\pm)}. \tag{D3}$$

which are the formulae we use in the following. In particular, we have $\hat{\mathcal{R}}^2(A) = \overline{(U \otimes U)(A \otimes A)(U^\dagger \otimes U^\dagger)}$

In the following, we will use the following identity, due to the cyclicity of the trace and the properties of a commutator

$$\text{Tr}([A, [B, C]]D) = \text{Tr}([[D, A], B]]C) = \text{Tr}([B, [A, D]]C), \tag{D4}$$

which can be derived in a few steps of cyclicity of the trace, and the last from flipping the commutators twice. This expression will be useful when averaging with twirls over the interaction $V_S$, as we can write, if $A = B$, then

$$\text{Tr}([H_0, [H_0, V_*]]\rho) = \text{Tr}([H_0, [H_0, \rho]]V_*) \tag{D5}$$

which an the expression we use in the following.

### Quantum battery

Let us now write this expression term by term. First, we note that $|\text{Tr}(A)|^2 = \text{Tr}(A)\text{Tr}(A)^* = \text{Tr}(A)\text{Tr}(A^\dagger)$, and $[A, B]^\dagger = [B^\dagger, A^\dagger]$. For the commutator term, we have for hermitian $A, B$

$$\left|\frac{1}{2i}\langle[A, B]\rangle\right|^2 = \frac{1}{4}\text{Tr}(\rho_t[A, B])\text{Tr}\left([A, B]^\dagger \rho_t^\dagger\right) \tag{D6a}$$

$$= \frac{1}{4}\text{Tr}\left([A, B]\rho_t \otimes [B^\dagger, A^\dagger]\rho_t\right) \tag{D6b}$$

$$= \frac{1}{4}\text{Tr}\left(([A, B] \otimes [B^\dagger, A^\dagger])(\rho_t \otimes \rho_t)\right) \tag{D6c}$$

$$= \frac{1}{4}\text{Tr}\left(([A, B] \otimes [B^\dagger, A^\dagger])_{-t}(\rho_0 \otimes \rho_0)\right) \tag{D6d}$$

which is the expression we use in the following.

We assume a random initial state as a twirl of the form

$$\boldsymbol{\rho}_0^U = U\boldsymbol{\rho}_0 U^\dagger \tag{D7}$$

It follows that

$$\overline{\left|\frac{1}{2i}\langle[\boldsymbol{A},\boldsymbol{B}]\rangle\right|^2} = \frac{1}{4}\operatorname{Tr}\left(([\boldsymbol{A},\boldsymbol{B}]\otimes[\boldsymbol{B}^\dagger,\boldsymbol{A}^\dagger])_{-t}(\lambda_+(\boldsymbol{\rho_0}^{\otimes 2})\boldsymbol{\Pi}_+ + \lambda_-(\boldsymbol{\rho_0}^{\otimes 2})\boldsymbol{\Pi}_-)\right) \tag{D8}$$

We note that we have

$$\operatorname{Tr}(\boldsymbol{\Pi}_\pm) = \frac{1}{2}\left(\operatorname{Tr}(\mathbb{I}) \pm \operatorname{Tr}(\mathbb{S})\right) = \frac{d_S(d_S \pm 1)}{2} \tag{D9}$$

and thus

$$\lambda_\pm(\boldsymbol{\rho_0}^{\otimes 2}) = \frac{\operatorname{Tr}(\boldsymbol{\rho_0})^2 \pm \operatorname{Tr}(\boldsymbol{\rho_0^2})}{d_S(d_S \pm 1)} = \frac{1 \pm \mathcal{P}}{d_S(d_S \pm 1)} \tag{D10}$$

where $\mathcal{P}$ is the purity of the initial state,

$$\mathcal{P} = \operatorname{tr}(\boldsymbol{\rho_0^2}). \tag{D11}$$

We then have

$$\overline{\left|\frac{1}{2i}\langle[\boldsymbol{A},\boldsymbol{B}]\rangle\right|^2} = \frac{1}{4}\operatorname{Tr}\left(([\boldsymbol{A},\boldsymbol{B}]\otimes[\boldsymbol{B}^\dagger,\boldsymbol{A}^\dagger])_{-t}\left((1+\mathcal{P})\frac{\boldsymbol{\Pi}_+}{d_S(d_S+1)} + (1-\mathcal{P})\frac{\boldsymbol{\Pi}_-}{d_S(d_S-1)}\right)\right) \tag{D12}$$

where $_t$ indicates that $U(t)[\boldsymbol{A},\boldsymbol{B}]U^\dagger(t)$ are time evolved individually.

Using the fact that

$$a\boldsymbol{\Pi}_+ + b\boldsymbol{\Pi}_- = \frac{a+b}{2}\mathbb{I} + \frac{a-b}{2}\mathbb{S} = l_i\mathbb{I} + l_s\mathbb{S} \tag{D13}$$

then we can rewrite the expression above as

$$\overline{\left|\frac{1}{2i}\langle[\boldsymbol{A},\boldsymbol{B}]\rangle\right|^2} = \frac{l_i}{4}\operatorname{Tr}([\boldsymbol{A},\boldsymbol{B}])\operatorname{Tr}([\boldsymbol{B}^\dagger,\boldsymbol{A}^\dagger]) + \frac{l_s}{4}\operatorname{Tr}([\boldsymbol{A},\boldsymbol{B}][\boldsymbol{B}^\dagger,\boldsymbol{A}^\dagger]). \tag{D14}$$

Note however that we can reduce the expression above simply to

$$\overline{\left|\frac{1}{2i}\langle[\boldsymbol{A},\boldsymbol{B}]\rangle\right|^2} = \frac{l_s}{4}\operatorname{Tr}([\boldsymbol{A},\boldsymbol{B}][\boldsymbol{B}^\dagger,\boldsymbol{A}^\dagger]) \tag{D15}$$

where we used the fact the trace of a commutator is zero, and where

$$l_s = \frac{d_S\mathcal{P} - 1}{d_S(d_S^2 - 1)}.$$

It is interesting to note that if $\mathcal{P} = 1/d_s$, then the right-hand side becomes zero. In the limit $d_S \gg 1$, we can then use the approximation

$$\left((1+\mathcal{P})\frac{\boldsymbol{\Pi}_+}{d_S(d_S+1)} + (1-\mathcal{P})\frac{\boldsymbol{\Pi}_-}{d_S(d_S-1)}\right) \approx_{d_S \gg 1} \frac{1}{d_S^2}\left(\mathbb{I} + \mathcal{P}\mathbb{S}\right). \tag{D16}$$

and obtain the formula

$$\overline{\left|\frac{1}{2i}\langle[\boldsymbol{A},\boldsymbol{B}]\rangle\right|^2} = \frac{\mathcal{P}}{(2d_S)^2}\operatorname{Tr}([\boldsymbol{A},\boldsymbol{B}][\boldsymbol{B}^\dagger,\boldsymbol{A}^\dagger]) \tag{D17}$$

which is the formula we will use in the following.

*Closed battery*

Let us now use the formula above in the case of the closed quantum system, e.g. the quantum battery. We replace $\boldsymbol{B} = -\frac{i}{\hbar}[\boldsymbol{H}_0, \boldsymbol{V}_S]$, and $\boldsymbol{A} = \boldsymbol{H}_0$. We have, averaging over the initial conditions, and noticing that $[\boldsymbol{H}_0, [\boldsymbol{H}_0, \boldsymbol{V}_S]]^\dagger = [\boldsymbol{H}_0, [\boldsymbol{H}_0, \boldsymbol{V}_S]]$, we have

$$\overline{\mathcal{B}}_\rho = \overline{\left| \frac{1}{2i\hbar} \langle [\boldsymbol{H}_0, [\boldsymbol{H}_0, \boldsymbol{V}_S]] \rangle \right|^2} = \frac{l_s}{(2\hbar)^2} \operatorname{Tr}\big([\boldsymbol{H}_0, [\boldsymbol{H}_0, \boldsymbol{V}_S]]^2\big) \tag{D18}$$

Using the properties of traces, and after straightforward calculation, it is easy to see that

$$\operatorname{Tr}\big([\boldsymbol{H}_0, [\boldsymbol{H}_0, \boldsymbol{V}_S]]^2\big) = \operatorname{Tr}\left( \boldsymbol{H}_0^2 \big( 6\boldsymbol{V}_S \boldsymbol{H}_0^2 \boldsymbol{V}_S - 8(\boldsymbol{H}_0 \boldsymbol{V}_S)^2 + 2\boldsymbol{H}_0^2 \boldsymbol{V}_S^2 \big) \right) \tag{D19}$$

which is the expression we use in the main text.

*Twirling the interaction*

We now consider the twirl of the interaction potential $\boldsymbol{V}_S$, similarly to what done in Ref. [79]. Using the expression for hermitian $\boldsymbol{H}_0, \boldsymbol{V}_S$,

$$\mathcal{B} = \frac{1}{4} \operatorname{Tr}\left( \big([\boldsymbol{H}_0, [\boldsymbol{H}_0, \boldsymbol{V}_S]] \otimes [[\boldsymbol{V}_S^\dagger, \boldsymbol{H}_0^\dagger], \boldsymbol{H}_0^\dagger]\big)(\boldsymbol{\rho}_t \otimes \boldsymbol{\rho}_t) \right) \tag{D20a}$$

$$= \frac{1}{4} \operatorname{Tr}\left( \big([\boldsymbol{H}_0, [\boldsymbol{H}_0, \boldsymbol{\rho}_t]] \otimes [\boldsymbol{H}_0, [\boldsymbol{H}_0, \boldsymbol{\rho}_t]]\big)(\boldsymbol{V}_S \otimes \boldsymbol{V}_S) \right) \tag{D20b}$$

in which we used expression (D5). Since this is exactly the same expression as before, with $\boldsymbol{\rho} \leftrightarrow \boldsymbol{V}_S$, we have that averaging over the unitary channel $U$ with $U^\dagger \boldsymbol{V}_S U$, we have

$$\overline{\mathcal{B}}_V = \frac{m_s}{(2\hbar)^2} \operatorname{Tr}\big([\boldsymbol{H}_0, [\boldsymbol{H}_0, \boldsymbol{\rho}_t]]^2\big) \tag{D21}$$

where

$$m_s = \frac{d_S \mathcal{V} - 1}{d_S(d_S^2 - 1)}, \tag{D22}$$

with $\mathcal{V} = \operatorname{tr}\big(\boldsymbol{V}_S^2\big)$, and also we can immediately write

$$\operatorname{Tr}\big([\boldsymbol{H}_0, [\boldsymbol{H}_0, \boldsymbol{\rho}_t]]^2\big) = \operatorname{Tr}\left( \boldsymbol{H}_0^2 \big( 6\boldsymbol{\rho}_t \boldsymbol{H}_0^2 \boldsymbol{\rho}_t - 8(\boldsymbol{H}_0 \boldsymbol{\rho}_t)^2 + 2\boldsymbol{H}_0^2 \boldsymbol{\rho}_t^2 \big) \right). \tag{D23}$$

Thus, unlike the twirling over the initial state, we have that $\overline{\mathcal{B}}_V$ is still time dependent.

**Bath-System interactions**

Let us now consider the average of the right-hand side of the commutator, with a bit more structure. We assume that $\boldsymbol{B} = [\boldsymbol{B}_0, [\boldsymbol{B}_0, \boldsymbol{C}_S \otimes \boldsymbol{C}_E]]$. We assume now that $\boldsymbol{C}_E \to U\boldsymbol{C}_E U^\dagger$.

Now let us assume that our operator $\boldsymbol{D} = \boldsymbol{D}_S \otimes \boldsymbol{D}_E$, $\boldsymbol{A} = \boldsymbol{B} = \boldsymbol{H}_0 \otimes \mathbf{1}_E$, and $\boldsymbol{C} = \boldsymbol{C}_S \otimes \boldsymbol{C}_E$. It is easy to see then that we can write, following eqn. (D4):

$$\operatorname{Tr}([[\boldsymbol{D}, \boldsymbol{A}], \boldsymbol{B}]\boldsymbol{C}) = \operatorname{Tr}_{SE}([[\boldsymbol{D}_S, \boldsymbol{H}_0], \boldsymbol{H}_0]\boldsymbol{C}_S \otimes \boldsymbol{D}_E \boldsymbol{C}_E) = \operatorname{Tr}_S([[\boldsymbol{D}_S, \boldsymbol{H}_0], \boldsymbol{H}_0]\boldsymbol{C}_S) \operatorname{Tr}_E(\boldsymbol{D}_E \boldsymbol{C}_E) \tag{D24}$$

In the equation above, we kept the notation to make clear that the traces are picked on different subspaces, but we will remove this extra notation in what follows. We can see that the right hand side of the uncertainty relationship is the case of the closed quantum battery multiplied by a factor associated with the environment, which we call $\mathcal{D}_E$.

We choose $\boldsymbol{D}_S = \boldsymbol{\rho}_S(t)$, while $\boldsymbol{D}_E = \boldsymbol{\rho}_E$ is thermal, and we assume time-independent. Similarly, the interaction is given by $\boldsymbol{C} = \boldsymbol{V}_S \otimes \boldsymbol{V}_E$.

*Twirling the system initial state*

Let us first note that twirling the initial density matrix in this setup is similar to the analysis performed in the previous section. We have

$$\overline{\mathcal{B}_\rho} = \overline{\left| \frac{1}{2i} \langle [\boldsymbol{A}, \boldsymbol{B}] \rangle \right|^2} = \frac{1}{4} \overline{\mathrm{Tr}\left( [\boldsymbol{H_0}, [\boldsymbol{H_0}, \boldsymbol{V}_S]] U_t^\dagger U^\dagger \boldsymbol{\rho}_{0S} U U_t \right)^2} \mathrm{Tr}\left( \boldsymbol{\rho}_E \boldsymbol{V}_E \right)^2 \tag{D25a}$$

$$= \frac{l_s}{(2\hbar)^2} \mathrm{Tr}\left( [\boldsymbol{H_0}, [\boldsymbol{H_0}, \boldsymbol{V}_S]]^2 \right) \mathrm{Tr}\left( \boldsymbol{\rho}_E \boldsymbol{V}_E \right)^2 \tag{D25b}$$

which is similar to the expression we had before, but now a factor due to the average of the interaction with the bath enters. We note that $\mathrm{Tr}\left( [\boldsymbol{H_0}, [\boldsymbol{H_0}, \boldsymbol{V}_S]]^2 \right) = \| [\boldsymbol{H_0}, [\boldsymbol{H_0}, \boldsymbol{V}_S]] \|_F^2$.

*Twirling the interacting potential*

We now assume that the unitary channel operates as $\boldsymbol{V}_E \to U \boldsymbol{V}_E U^\dagger$.

$$\overline{\mathcal{B}_V} = \overline{\left| \frac{1}{2i} \langle [\boldsymbol{A}, \boldsymbol{B}] \rangle \right|^2} = \frac{1}{4} | \mathrm{Tr}\left( [\boldsymbol{H_0}, [\boldsymbol{H_0}, \boldsymbol{V}_S]] \boldsymbol{\rho}_S(t) \right) |^2 \cdot \overline{| \mathrm{Tr}\left( \boldsymbol{\rho}_E U \boldsymbol{V}_E U^\dagger \right) |^2} \tag{D26}$$

$$= \frac{\overline{\mathcal{M}_E}}{4} | \mathrm{Tr}\left( [\boldsymbol{H_0}, [\boldsymbol{H_0}, \boldsymbol{V}_S]] \boldsymbol{\rho}_S(t) \right) |^2 \tag{D27}$$

We see from the equation above that the uncertainty is the same as the one of a quantum battery, but with a factor that depends on the bath $\mathcal{M}_E$. It is interesting to note that, because of the invariance of the Haar measure, and/or because of the ciclity of the trace, this is the same as averaging via $\boldsymbol{\rho}_E \to U^\dagger \boldsymbol{\rho}_E U$, while keeping $\boldsymbol{V}_E$ constant. For the average of this term, we can use the formulae from the previous section, to obtain

$$\overline{\mathcal{M}_E} = \overline{\mathrm{Tr}\left( \boldsymbol{\rho}_E U \boldsymbol{V}_E U^\dagger \right)^2} = \frac{\mathrm{Tr}\left( \boldsymbol{V}_S^{\otimes 2} \boldsymbol{\Pi}^+ \right)(1 + \mathcal{P}_E)}{d_E(d_E + 1)} + \frac{\mathrm{Tr}\left( \boldsymbol{V}_S^{\otimes 2} \boldsymbol{\Pi}^+ \right)(1 - \mathcal{P}_E)}{d_E(d_E - 1)} \tag{D28}$$

where $\mathcal{P}_E = \frac{1}{Z^2} \mathrm{Tr}\left( e^{-2\beta \boldsymbol{H}_E} \right)$ is the purity of the bath. For $\beta = 0$, we have $\mathcal{P}_E = 1/d_E$, while for $\beta = \infty$, $\mathcal{P}_E = 1$. Note that in the limit of a large bath, we obtain, using the fact that $\boldsymbol{\Pi}^+ + \boldsymbol{\Pi}^- = \mathbb{I}_E$, and $\boldsymbol{\Pi}^+ - \boldsymbol{\Pi}^- = \mathbb{S}_E$,

$$\overline{\mathcal{M}_E} = \overline{\mathrm{Tr}\left( \boldsymbol{\rho}_E U \boldsymbol{V}_E U^\dagger \right)^2} \approx_{d_E \gg 1} \frac{1}{d_E^2} \left( \mathrm{Tr}\left( \boldsymbol{V}_E^{\otimes 2} \right) + \mathrm{Tr}\left( \boldsymbol{V}_E^2 \right) \mathcal{P}_E \right) = \frac{1}{d_E^2} \left( \mathrm{Tr}(\boldsymbol{V}_E)^2 + \mathrm{Tr}\left( \boldsymbol{V}_E^2 \right) \mathcal{P}_E \right) \tag{D29}$$

$$\tag{D30}$$

We thus obtain that depending on the interaction potential, the uncertainty is renormalized by an interaction with the bath and the temperature. If we define the variance of an operator in the infinite temperature bath, e.g.

$$\Delta V_\infty^2 = \frac{1}{d_E} \left( \mathrm{Tr}\left( \boldsymbol{V}_E^2 \right) - \mathrm{Tr}(\boldsymbol{V}_E)^2 \right), \tag{D31}$$

then we have

$$\overline{\mathcal{M}_E} = \frac{\mathrm{Tr}\left( \boldsymbol{V}_E^2 \right)(1 + \mathcal{P}_E) - d_E \Delta V_\infty^2}{d_E^2} \tag{D32}$$

where $\mathcal{P}_E$ is the purity of the bath.