# Peer review of "Operator-based quantum thermodynamic uncertainty relations"

_SciPost Physics_

## Round 2 · Referee Report · Philipp Strasberg (Referee 1) · 2024-10-29

Report

In my view, the compelling features of the thermodynamic uncertainty relation (TURs) in Eq. (1) are that they bound (A) the signal-to-noise ratio of physical quantities (B) in terms of the entropy production (C) in a simple way. If I see things correctly, these desirable features are not met in the present paper:

(A) The bounds feature the variance but not the signal-to-noise ratio (mean value over variance), which seems undesirable as the right comparative scale is missing.
(B) The bounds do not contain the entropy production.
(C) The bounds, while being very (or even completely) general, seem rather involved once they are applied to some system. Indeed, the authors restrict examples to the simplest cases (and even there encounter problems, see Fig. 7b) and the time-dependence of the bound seems more complicated than the quantity they want to bound (Figs. 3 and 4).

To be fair, the authors emphasize that their "operator TURs" should not be directly compared with traditional TURs, but any compelling reason why the operator TUR is relevant and insightful is missing as far as I can see. The Robertson-Schroedinger uncertainty relation is completely general and I can apply it to any pair of operators. What's the point? To me, the physical meaning of the variance of the operators in, e.g., Eq. (17) even remains unclear. Moreover, the bound sometimes seems to be rather useless (Fig. 6b).

To conclude, I do not think that the paper is suitable for SciPostPhysics. It is certainly well written (even though some calculations are a bit too repetitive and detailed for my taste) and the results seem a priori novel. Therefore, it could be an option for SciPostPhysics Core.

Finally, I would ask the authors to point out that the identification in Eq. (6) might only hold in the weak coupling regime.

Recommendation

Reject

---

## Round 2 · Referee Report · Anonymous (Referee 2) · 2024-11-11

Strengths

1) Timely topic addressing fluctuations in thermodynamic settings which are quantum 2) Well structured manuscript 3) Comprehensive study of examples

Weaknesses

1) (Key criticism) Misleading analogy with the 'thermodynamic uncertainty relations', details in the report.

Report

In their work titled "Operator-based quantum thermodynamic uncertainty relations", the authors address the timely topic of thermodynamic uncertainty relations (TUR) from the perspective of quantum mechanical operators. In the abstract and introduction they announce that they derive a fluctuation bound for quantum mechanical operators akin to the 'current' TUR known in classical stochastic systems where for a current $J$ the bound $\langle\langle J^2\rangle\rangle / \langle J\rangle^2 \geq 2/\Sigma$ holds with $\Sigma$ the entropy production rate. This topic has had a surge in interest in the last years from the community working on classical statistical mechanics but also from the quantum thermodynamic community. One of the open challenges in this field is how the classical TUR can be extended for quantum systems in a form similar to the TUR where fluctuations stand in a trade-off against entropy production. So far, only in some restrictive settings, such quantum bounds have been shown, and this work claims to approach this challenge from the perspective of bounding operator variances. In the main text, the authors then show refer to the well-known Robertson-Schrödinger inequality for operators and use this as the basis for their operator TUR. For this, they consider a system+environment setting where they define work and heat operators and then they consider the variances of such operators. They consider this in a global unitary picture and in a quantum master equation. Their main part is rounded off by two key examples they study: interacting spins and harmonic oscillators, and the authors show how their bound performs in these settings. In the second part of their work, the authors consider the example of quantum batteries and they explore what their operator TUR says about fluctuations in battery energy and charging power. Finally, they conclude with a typicality analysis of their bound.

Before I go into the detail of the report, I would like to address my main point of criticism and the reason why I am against the publication of this work in it's current form: The work claims to prove/find a bound akin to the TUR but for quantum operators. Unfortunately, this is not the case. Let me reiterate: the TUR bound fluctuations by entropy production and those bounds are interesting because in many settings we care about low entropy production and low fluctuations. The TUR now inform us that this is not possible. There are also the kinetic uncertainty relations (KUR) which have a quantity called dynamical activity instead of entropy production. The present work, however, does not bound fluctuations in terms of entropy production or activity, but in terms of (anti)commutators of the operators whose fluctuations are examined. It is therefore, in my opinion, highly misinformative if the bound is called an 'operator based TUR'. I therefore strongly urge the authors to reformulate their title, abstract and introduction to clarify the context of their bounds.

Going a bit more into the detail of what appears to be their main result, the operator TUR (e.g. eq. (2) or (27)) in it's general form is the Robertson-Schrödinger (RS) inequality (what they cite around eq. (10)) applied to the power operator and heat flow operator. This is of course not a problem, but in this general form, this can hardly be considered a operator TUR. It is simply the standard RS uncertainty relation for power and heat operators, but it has a priori nothing to do with the TUR. As written before, this is my main point of critique, and if the authors manage to sufficiently reformulate how their result is a fluctuation bound for power/heat (and the other quantities they consider) without confusing it with the TUR, I would generally be happy to reconsider this work for publication.

In the following, I give some more detailed comments partially on the topic of the issue raised above, and on other smaller things I saw:

  1. The literature of the TUR initially is not ideally structured. First, the authors talk about 'classical stothastic thermodynamics' and cite the Nat. Phys. perspective [14]. Here, the original TUR work would be more appropriate in my opinion. Then, however, when they cite Refs. [15-22] there are both classical works (including the original ones by Barato and Seifert) but also ones by Hasegawa et al on quantum versions of the TUR. I suggest the authors to structure their literature more comprehensively: clearly clarify what are the results in classical stochastic systems and what are the results for quantum systems. It doesn't have to encompass all the literature, of course, and does not need to be a review. A concise but well-structured account of past work in this area would be sufficient. (Comment: given that their bound is not actually a TUR, this part may not be needed at all, after all)
  2. The statement on p.2 that TUR are 'usually applicable only to NESSs' is not true, finite time versions have been proven.
  3. I think the paper is well-organized, with good overview on p.2 and a very comprehensive discussion of the setup.
  4. Maybe somewhere along p.3,4 or 5 the authors should discuss the work by Nishiyama and Hasegawa (https://iopscience.iop.org/article/10.1088/1751-8121/ad79cd) titled 'Tradeoff relations in open quantum dynamics via Robertson, Maccone–Pati, and Robertson–Schrödinger uncertainty relations'. The title I think makes it clear this is a highly relevant reference for the present work, however, I believe the two are of course sufficiently different. (Comment: there, they also derive what they call TURs and actually the same criticism I am raising for the present work applies there too.)
  5. Another comment that is not related to a specific location of the manuscript, but may potentially be of interest to the authors: In the Nat. Comm by Hasegawa (https://www.nature.com/articles/s41467-023-38074-8), it is shown how TURs and Uncertainty relations can be unified formally using continuous matrix product states. But other than this superficial relation, the work is not too relevant for the present manuscript, I am simply flagging this, as a potentially interesting reference for the authors.
  6. After eq. (8), the authors may point toward the place in their manuscript where they talk in more detail about the quantum master equation, or they may want to state that the quantum master equation only holds in some restrictive settings (with appropriate reference).
  7. Important: In their Sec. III B the authors (as far as I can tell) do not discuss ambiguities to the definition of heat and work in open quantum systems beyond the weak coupling regime. I am aware these are standard assumptions, but the quantum thermodynamics community should not collectively forget these (restrictive) assumptions made in order to have well-behaved notions of heat etc. Thus, I urge the authors to clearly state these caveats about interpreting the physical quantities like heat and power, and moreover, I suggest the authors to clearly mention the assumptions necessary to make such quantities well-defined. Personally, I like the reference by Landi and Paternostro (https://journals.aps.org/rmp/abstract/10.1103/RevModPhys.93.035008) but the authors should decide themselves how to adress this point.
  8. As for the examples (general comment), I think they are quite interesting case studies, and the results that come out, as for example in eq. (33b) are sufficiently concise uncertainty relations that have some physical meaning. This is what I would consider as a 'result', not eq. (27). At the same time, it is still not a TUR, rather, an operator uncertainty relation, and at most it bears resemblence with the KUR (where in some limit the term $\langle V_{\rm SE}\rangle$ can be considered related to the dynamical activity).
  9. General comments about Figs. 3-5: Those seem to confirm the theory calculations which is of course good, but I am missing a bit the discussion of the physics going on in these examples. The authors are writing about heat and power, thus a comment on the physical interpretation would be a minimum.
  10. I generally like the study of the quantum battery example, which is I believe a relevant case study here.
  11. The typicality calculations at the end also look interesting as they yield some results not depending on the instantaneous state. This is a good thing to consider, though unfortunately, the final result does not look much more instructive than the time-dependent quantity obtained previously.

In summary, my impression of the author's work is that they provide consider standard uncertainty relations for operators with a thermodynamic meaning like that of power, heat etc. Using the Robertson-Schrödinger inequality, they exercise through an extensive set of examples and provide numerical data showcasing their result. Seen like this, it is a useful but technical contribution. At the same time, it does not meet the 'acceptance criteria' the authors selected: a) Open a new pathway in an existing or a new research direction, with clear potential for multi-pronged follow-up work b) Detail a groundbreaking theoretical/experimental/computational discovery The reasons: a) TURs in the quantum real have already been studied, the RS inequality has already been applied to TUR-like settings to look at fluctuations of thermodynamic quantities and fluctuation bounds have also been considered already. Thus I do not believe the authors have opened up a new research direction. b) Also the result is not a groundbreaking discovery. Surely, most works are not 'truly' groundbreaking, so I don't think this should be the criteria, but discovery should be a criteria. All-in-all, I believe the authors are showcasing and analyzing examples, which is useful and a valuable scientific contribution, but it is not a discovery.

I thus believe that this paper belongs to a less prestigeous journal of Sci|Post than SciPost Physics. Still, I believe the publication should be conditioned on how the main result is presented in the context of the TURs because as of now, title, abstract and introduction are misleading. If this is fundamentally changed, I am happy having a new look at this work. Despite this review being quite critical, the example calculations seem to be well executed and I thus think the content of this work can generally be a valuable contribution to the scientific literature.

Requested changes

Please find all suggested changes in the report.

Recommendation

Reject

---

## Editorial Decision

awaiting_resubmission